# Effects of Volume Changes on the Thermal Performance of PCM Layers Subjected to Oscillations of the Ambient Temperature: Transient and Steady Periodic Regimes

**DOI:** 10.3390/molecules27072158

**Published:** 2022-03-27

**Authors:** Rubén D. Santiago-Acosta, Ernesto M. Hernández-Cooper, Rolando Pérez-Álvarez, José A. Otero

**Affiliations:** 1Tecnologico de Monterrey, Escuela de Ingeniería y Ciencias, Carr. al Lago de Guadalupe Km. 3.5, Atizapán de Zaragoza 52926, Mexico; ruben.dario@tec.mx (R.D.S.-A.); emcooper@tec.mx (E.M.H.-C.); 2Centro de Investigación en Ciencias, Universidad Autónoma del Estado de Morelos, Cuernavaca 52900, Mexico; rpa@uaem.mx

**Keywords:** several front formation, phase change material, thermal performance

## Abstract

The Stefan problem regarding the formation of several liquid–solid interfaces produced by the oscillations of the ambient temperature around the melting point of a phase change material has been addressed by several authors. Numerical and semi-analytical methods have been used to find the thermal response of a phase change material under these type of boundary conditions. However, volume changes produced by the moving fronts and their effects on the thermal performance of phase change materials have not been addressed. In this work, volume changes are incorporated through an additional equation of motion for the thickness of the system. The thickness of the phase change material becomes a dynamic variable of motion by imposing total mass conservation. The modified equation of motion for each interface is obtained by coupling mass conservation with a local energy–mass balance at each front. The dynamics of liquid–solid interface configurations is analyzed in the transient and steady periodic regimes. Finite element and heat balance integral methods are used to verify the consistency of the solutions to the proposed model. The heat balance integral method is modified and adapted to find approximate solutions when two fronts collide, and the temperature profiles are not smooth. Volumetric corrections to the sensible and latent heat released (absorbed) are introduced during front formation, annihilation, and in the presence of two fronts. Finally, the thermal energy released by the interior surface is estimated through the proposed model and compared with the solutions obtained through models proposed by other authors.

## 1. Introduction

Liquid–solid phase transitions have industrial applications, where on the one hand, phase change materials (PCMs) can be used as backup systems for thermoelectric generation [1,2] and on the other hand, as thermal barriers for air conditioning applications in the building industry [3,4]. The energy density of materials is enhanced through the latent heat absorbed in thermal energy storage units for thermoelectric generation during periods of low solar irradiance [5]. In contrast to applications related to thermoelectric generation, PCMs with low thermal conductivities are highly desirable when used as a thermal barrier to provide thermal comfort in building applications. The performance of PCMs used as thermal shields through PCM wallboards [4,6], PCM ceilings [7] and energy storage units for day/night cooling to reduce energy consumption [8], has been previously studied. Numerical simulations have been used to analyze the thermal performance of PCM walls, where the exterior surface is subjected to daily periodic boundary conditions and under real weather data [9]. The effects of natural convection on PCM layers with different orientations have been previously studied [10]. The authors conclude that energy transfer rates are reduced in PCM ceilings when the exterior surface is above the melting temperature $T_{m}$ and the liquid phase is in the conductive regime.

Numerical methods have been used to incorporate volume displacements produced by the density change during the phase transition [11,12,13,14]. The authors of Refs. [11,12] use several ways to accommodate mass during freezing of liquid water in confined systems. The authors perform numerical simulations with models that conserve mass properly, while considering the density difference between ice and liquid water. They found that solidification times are 19% higher and the energy absorbed by the PCM is 9% higher, when compared with models where the total mass of the system is not conserved. Finally, the authors assume incompressible phases, so the methods proposed can only be applied for isobaric transitions or materials with low isothermal compressibility. Thermo-mechanical models that consider the volume changes of a melting salt when confined in an elastic spherical shell have been developed [15]. The effects of the salt volume expansion played a significant role on the melting dynamics, thermal energy stored, and mechanical response of the shell. Experimental estimations of the latent heat, sensible heat, and mechanical energy stored in a micro-encapsulated ${\mathrm{KNO}}_{3}/{\mathrm{NaNO}}_{3}$ salt were performed [16]. The thermo-mechanical model proposed in Ref. [15] was tested against the experimental estimations of the melting dynamics [16]. The results show a qualitative agreement between the the proposed model and the experimental estimations. The observed difference can be attributed to density changes produced by the thermal stress during the melting process and that are not considered in the original model [15]. Experimental studies of micro-encapsulated phase change materials (MEPCMs) with voids have also been performed [17]. The voids prevent the loss of latent heat storage capacity produced by the thermal stress during the melting process. Density variations produced by the thermal stress during melting of confined salts have also been studied. The model proposes a mass balance that considers the compressibility of each phase [18], extending the applicability of the original models [11,12,15] to the melting of confined PCMs.

Semi-analytical solutions to the Stefan problem on finite systems and with different types of boundary conditions have been reported in Refs. [19,20]. Periodic boundary conditions were preferred, since PCM layers used for thermal shielding applications are subjected to temperature oscillations produced by daily thermal variations [20]. The thermal performance of PCM walls and ceilings for building applications with periodic boundary conditions has also been addressed. Semi-analytical solutions to the Stefan problem in a finite PCM layer subjected to periodic boundary conditions have been previously reported [20]. The problem is simplified by assuming temperature oscillations on the external surface that are always above the melting temperature of the PCM. Additionally, temperature variations on the interior surface and below the melting temperature of the PCM were assumed. The situation produces the formation of one liquid-solid interface in the PCM layer that oscillates with the same frequency as the thermal oscillations on the layer boundaries [20]. The problem was also addressed in Ref. [21], where the authors use an enthalpy method and neglect volume changes produced by the oscillations of the interface. Experimental and numerical evidence in PCM layers subjected to temperature oscillations above the melting temperature on the external surface has been reported [22]. The temperature on the interior surface was fixed to a constant value and below the melting temperature of the PCM through a cooling system. Good agreement between the experimental results and a model where volume changes were not considered, was observed in the liquid phase. Discrepancies between the numerical predictions and the experimental results were observed on the temperature oscillations in the solid phase. According to the authors, the observed discrepancies could be produced by the assumption of one dimensional heat transfer. The weather or experimental conditions may produce the formation of several liquid–solid interfaces or fronts. Temperature oscillations around the melting temperature of the PCM layer on the external surface and homogeneous temperatures below the melting point of the PCM on the internal surface, have also been considered [23]. The authors found that during the heating process, the solid PCM melts on the exterior surface and the interface or front moves towards the interior surface. Additionally, when the ambient temperature is below $T_{m}$, a thin solid layer in contact with the external surface is formed. The PCM layer is then divided into three regions and in the presence of two fronts during the cooling process. Formation of three liquid-solid interfaces may also occur, dividing the PCM layer into four regions. The configuration of phases and number of existing fronts depends on the amplitude of the temperature oscillations, the thickness of the PCM layer and thermodynamic properties of the PCM [23]. Periodical and non-sinusoidal boundary conditions have also been considered to address the effects of weather variations due to season changes during an entire year [24,25]. The phase configuration in the PCM layer when the exterior surface is subjected to variable periodic boundary conditions may change during each season. The Stefan problem with periodic boundary conditions around or above the melting temperature of the PCM, has been extensively studied. The problem studied by the mentioned authors has a one-dimensional treatment and no volume changes are incorporated. The literature that is focused on the Stefan problem with applications on thermal shielding, frequently neglects volume changes when describing the phase transition dynamics. It is commonly stated that volume changes in liquid-solid transitions represent at most 10% of the total volume of the system and can be neglected in the modeling by assuming phases with equal densities [26]. Recently, the issue of volume changes during isobaric transitions and in several types of organic PCMs was addressed to test this claim [26]. The authors found that, depending on the chemical composition of the PCM, volume changes can represent as much as 24% of the original volume in polymers. According to the authors, the treatment of volume changes experienced by the PCM during the phase transition represents a challenge that needs to be addressed in practical applications.

The effects of volume change on the thermal performance of PCM layers subjected to temperature oscillations about the melting temperature of the PCM has not been addressed. In this work, we consider the effects of volume changes on the energy released (absorbed) and energy transferred by the PCM in the presence two fronts. The energy transferred by the PCM is significantly influenced by the volume changes of the system and mainly in the presence of the two-front configuration, during the cooling stage of the cycle. A model that incorporates volume changes by promoting the layer thickness to a dynamic variable, is proposed. The oscillations of the layer thickness in the presence of several fronts obey an additional equation of motion that results from imposing total mass as a constant of the motion. The equation of motion for each interface is obtained through a local energy–mass balance that must be consistent with conservation of mass. The density used in each equation of motion results from the mass balance proposed in this work. The thermal performance of the PCM layer in the transient and steady periodic regimes, is analyzed through the thermal energy released (absorbed). The volumetric effects on the latent heat and sensible heat released (absorbed) during the solidification (melting) process are discussed in detail. Numerical and semi-analytical methods consisting on the finite element method (FEM) and heat balance integral method (HBIM), are used to solve the proposed equations of motion. Two different methods are used to verify the consistency of the solutions for the dynamical variables, released (absorbed) energy by the PCM layer and the energy released by the interior surface. The collision of two fronts produces a continuous temperature profile that is not smooth at the collision site. The HBIM is adapted by introducing a local energy balance at the collision site and predicting the time evolution of the temperature field just after the annihilation of two interfaces. The FEM is used to verify the consistency of the semi-analytical solutions. The total mass of the PCM layer is registered at every time interval to guarantee that no mass is created or destroyed during the entire phase change process. The time evolution of the released (absorbed) sensible and latent heats, and the energy released by the interior surface into the room are compared with models proposed by other authors. Finally, we find that the efficiency of a PCM layer as a thermal barrier close to the steady periodic regime may be reduced when considering the volume changes during the phase transition.

## 2. Description of the Physical System and Mathematical Model

The volumetric effects produced by the density difference between liquid and solid phases is considered when the external surface is subjected to periodic boundary conditions and the temperature of the internal surface is constant and below $T_{m}$. The temperature dependence of liquid and solid densities is negligible within the temperature range and PCM considered in this work [27]. Octadecane is used as the PCM, and within the experimental error, the density of the solid is practically constant in a wide temperature range [260,301.13]K. The largest variation of the liquid density is less than 1.5% within the maximum temperature range in the liquid phase [301.13,313.15]K, that will be used in this work. Thermal expansion effects are negligible and only volume variations produced by the density difference between liquid and solid phases will be considered. Two case scenarios will be discussed:Temperature oscillations on the external surface above the melting temperature of the PCM: one-front dynamics, andTemperature oscillations around the melting temperature of the PCM: two-front dynamics with three phase coexistence, one-front dynamics with two phase coexistence and no phase change presence.

The system under consideration consists of a PCM that constitutes a layer of thickness *L* and cross section *S* with a left (outer) boundary in contact with the ambient air and the right (inner) boundary in contact with the air at the interior of a room. The thermal flux is perpendicular to the surface of the PCM layer and the temperature is uniformly distributed throughout the exterior and interior surfaces. Isothermal boundary conditions are employed at each liquid–solid interface with a temperature value equal to the melting temperature $T_{m}$ of the PCM. The right boundary at x=L(t) is subjected to isothermal boundary conditions and the left boundary at the outside air–PCM interface (x=0) is subjected to periodic boundary conditions, as follows:(1)$$\begin{array}{ccc} T_{2}\left(0,t\right) & = & T_{0}+\delta \;\sin\left(\omega t+\phi \right)\;\mathrm{and},\\ T_{2}\left(\xi_{1}\left(t\right),t\right) & = & T_{1}\left(\xi_{1}\left(t\right),t\right)=T_{m},\\ T_{1}\left(L(t),t\right) & = & T_{C}, \end{array}$$
where $T_{0}$ represents the average daily temperature, δ the amplitude of the temperature oscillations, ω=2π/T the angular frequency with a 24h period and ϕ is the phase angle. The liquid solid front at any time *t* is located at $x=\xi_{1}\left(t\right)$. The oscillations of the ambient temperature were obtained through fitting weather data to a periodic function, in a tropical region located at Villahermosa, Tabasco in Mexico [28]. The temperature on the right boundary at x=L(t) is represented by $T_{C}$ and is below the melting temperature of the PCM. The temperature is uniform and homogeneously distributed throughout the interior surface. Maximum temperature differences in the liquid are $T_{H}-T_{m}\ll \;T_{m}$, and in the solid phase are $T_{m}-T_{C}\ll \;T_{m}$. The net heat flux at the interface is small in these temperature ranges and the phase transition takes place close to thermodynamic equilibrium; then, supercooling effects are not considered. Additionally, octadecane has a low supercooling degree within the operating temperature range considered in this work [29].

### 2.1. One-Front Dynamics: Transient and Steady Periodic Regime

The thickness of the PCM layer must incorporate volume changes during the phase transition according to total mass conservation and it is promoted to a dynamic variable defined as L(t). Volume changes have been previously considered during phase transitions at constant pressure [14]. We briefly describe the corresponding equations of motion, which can be applied to any type of boundary conditions. In a PCM layer of thickness L(t), when the temperature at x=0 is above $T_{m}$, the domain of the liquid phase lies on the interval: $0\leq \;x\leq \;\xi_{1}\left(t\right)$, where $\xi_{1}\left(t\right)$ is the position of the liquid-solid interface. Additionally, the domain of the solid phase lies on the interval $\xi_{1}\left(t\right)\leq \;x\leq \;L\left(t\right)$, where the boundary at x=L(t) is always below $T_{m}$. The local energy-mass balance at $x=\xi_{1}\left(t\right)$ that is consistent with volume displacements on these type of configurations [14], is given by:(2)$$\rho_{\ell }\;L_{f}\frac{d\;\xi_{1}\left(t\right)}{dt}=-k_{\ell }\frac{\partial \;T_{2}^{\left(\ell \right)}\left(x,t\right)}{\partial \;x}{|}_{x=\xi_{1}\left(t\right)}+k_{s}\frac{\partial \;T_{1}^{\left(s\right)}\left(x,t\right)}{\partial \;x}{|}_{x=\xi_{1}\left(t\right)},$$
where $T_{1}^{\left(s\right)}\left(x,t\right)$($T_{2}^{\left(\ell \right)}\left(x,t\right)$) is the temperature distribution in the solid(liquid) phase as illustrated in Figure 1, $\rho_{\ell }$ is the density of the liquid phase and $L_{f}$ is the latent heat of fusion of the PCM. The equation of motion for L(t) can be obtained straightforward through the total time derivative of the PCM’s mass per unit area *A*, and promoting this quantity as a constant of the motion, as follows:(3)$$\rho_{s}\;\frac{d\xi_{1}\left(t\right)}{dt}+\rho_{s}\;\left(\frac{d\;L(t)}{dt}-\frac{d\;\xi_{1}\left(t\right)}{dt}\right)=0.$$

Equations (Equation 2) and (Equation 3) are valid for melting and solidification processes, and consider volume changes during the phase transition produced by the density difference between liquid and solid. The density of the liquid phase appears in Equation (Equation 2) when the moving boundary is L(t), as shown in Ref. [14]. The local energy–mass balance within the liquid and solid phases is taken into consideration through the following heat equation:(4)$$\rho_{i}\;C_{i}\;\frac{\partial \;T_{i}\left(x,t\right)}{\partial \;t}-k_{i}\;\frac{\partial^{2}\;T_{i}\left(x,t\right)}{\partial \;x^{2}}=0,$$
where $\rho_{i}$, $C_{i}$ and $k_{i}$ is the density, specific heat capacity and thermal conductivity of phase *i*, with i=ℓ(s) for liquid (solid). The one-front dynamics problem is defined through Equations (Equation 2)–(Equation 4), with the corresponding boundary conditions, given by Equation (Equation 1).

Homogeneous isothermal boundary conditions have been applied to the one-front dynamics problem where, on the one hand, the temperature at x=0 is constant and higher than $T_{m}$. On the other hand, the temperature at x=L(t) is constant and below $T_{m}$. Steady state solutions for the thickness L(t) and liquid-solid interface position $\xi_{1}\left(t\right)$ have been found when the system is subjected to this type of boundary conditions [14]. Periodic boundary conditions are applied at x=0, which emulate the ambient temperature on the exterior surface of the PCM layer. The steady state values found in Ref. [14] can be applied to define upper and lower bounds to the interface position and layer thickness when the system is subjected to the boundary conditions shown in Equation (Equation 1). According to the steady state values for the liquid–solid interface position found in Ref. [14], an upper and lower bound for $\xi_{1}$ when the system is in the steady periodic regime and subjected to the boundary conditions given by Equation (Equation 1) are given by:(5)$$\xi_{sp}^{\left(u\right)}=\frac{k_{\ell }\left(T_{0}+\delta -T_{m}\right)\left(\rho_{s}L\left(0\right)-\left(\rho_{s}-\rho_{\ell }\right)\xi_{1}\left(0\right)\right)}{\rho_{\ell }\;k_{\ell }\left(T_{0}+\delta -T_{m}\right)+\rho_{s}\;k_{s}\left(T_{m}-T_{C}\right)},\;\mathrm{and}$$
(6)$$\xi_{sp}^{\left(l\right)}=\frac{k_{\ell }\left(T_{0}-\delta -T_{m}\right)\left(\rho_{s}L\left(0\right)-\left(\rho_{s}-\rho_{\ell }\right)\xi_{1}\left(0\right)\right)}{\rho_{\ell }\;k_{\ell }\left(T_{0}-\delta -T_{m}\right)+\rho_{s}\;k_{s}\left(T_{m}-T_{C}\right)},\;$$
where $\xi_{sp}^{\left(u\right)}$ and $\xi_{sp}^{\left(l\right)}$, represent the upper and lower bound of the interface position in the steady periodic regime. Highest and lowest temperature values are represented by $T_{0}+\delta$ and $T_{0}-\delta$. The initial interface position and thickness of the PCM layer are represented by $\xi_{1}\left(0\right)$ and L(0), respectively. Additionally, we can apply the analytical expression for the thickness of the PCM layer in the steady state [14], to determine the upper and lower bounds of *L* in the steady periodic regime as follows:(7)$$L_{sp}^{\left(u\right)}=\frac{\left(k_{s}\left(T_{m}-T_{C}\right)+k_{\ell }\left(T_{0}+\delta -T_{m}\right)\right)\left(\rho_{s}L\left(0\right)-\left(\rho_{s}-\rho_{\ell }\right)\xi_{1}\left(0\right)\right)}{\rho_{\ell }\;k_{\ell }\left(T_{0}+\delta -T_{m}\right)+\rho_{s}\;k_{s}\left(T_{m}-T_{C}\right)},\;\mathrm{and}$$
(8)$$L_{sp}^{\left(l\right)}=\frac{\left(k_{s}\left(T_{m}-T_{C}\right)+k_{\ell }\left(T_{0}-\delta -T_{m}\right)\right)\left(\rho_{s}L\left(0\right)-\left(\rho_{s}-\rho_{\ell }\right)\xi_{1}\left(0\right)\right)}{\rho_{\ell }\;k_{\ell }\left(T_{0}-\delta -T_{m}\right)+\rho_{s}\;k_{s}\left(T_{m}-T_{C}\right)}.$$
Here, $L_{sp}^{\left(u\right)}$($L_{sp}^{\left(l\right)}$) represents the upper(lower) bound of *L* in the steady periodic regime. Mass conservation in Equation (Equation 2) was applied by promoting the boundary in contact with the interior at x=L(t) to a dynamical variable. Consequently, Equations (Equation 5)–(Equation 8) are only valid when the exterior surface of the PCM layer at x=0 is fixed in time and the right boundary in contact with the interior at x=L(t) is the moving boundary.

### 2.2. Two-Front Dynamics

The formation of two or more liquid-solid fronts and coexistence of several adjacent liquid and solid phases, depends on the oscillations of the ambient temperature, the thermodynamic properties of the PCM and layer thickness. In this part of the section, the equations of motion that describe the effects of volume displacements on the dynamics of two liquid–solid interfaces will be introduced. The thermodynamic properties of the PCM, layer thickness and ambient temperature used in this work, will produce two liquid-solid fronts during the periods of the day when the ambient temperature is lower than $T_{m}$. The two fronts will collide at some instant during the cooling stage, and one liquid-solid front will be formed and evolve in time when the ambient temperature oscillates above $T_{m}$.

Initially, the configuration of the PCM layer will consist of two thin solid slabs and the ambient temperature is below $T_{m}$. The ambient temperature lies below the melting temperature of the PCM for the first few hours of the initial part of the cycle, when the two initial solid fronts propagate and collide during the cooling stage. The PCM layer will be in its solid state after the collision when only sensible heat is absorbed, while the ambient temperature is increasing but is still below $T_{m}$. The heating process starts when the ambient temperature is increasing and reaches $T_{m}$. A thin liquid slab is formed on the exterior surface and a single front dynamics is observed during the time interval where the ambient temperature is above $T_{m}$. The single front position is bounded by the value shown in Equation (Equation 5); therefore, a one-front configuration will always be present during the heating process of the cycle. The cooling process starts when the ambient temperature is decreasing and reaches $T_{m}$. Finally, a thin solid slab is formed on the exterior surface while the existing front within the PCM layer is moving towards the recently formed interface. The PCM layer is under the presence of a two-front configuration during the cooling stage, and the process is repeated.

Figure 2, is an schematic illustration of the front and phase configuration during the periods of the day where two liquid-solid fronts are present. The temperature distribution within each phase is labeled according to regions: 1, 2 and 3 from right to left, as shown in Figure 2. Regions 1, 2 and 3 represent a solid-liquid-solid phase configuration with a temperature distribution $T_{1}^{\left(s\right)}\left(x,t\right)$, $T_{2}^{\left(\ell \right)}\left(x,t\right)$ and $T_{3}^{\left(s\right)}\left(x,t\right)$, respectively. The temperature on the interior surface (right boundary) at x=L(t) is constant and equal to $T_{C}<\;T_{m}$. Figure 2 assumes that the temperature on the exterior surface is below the melting temperature of the PCM; therefore, the liquid layer between $\xi_{2}\left(t\right)$ and $\xi_{1}\left(t\right)$ will gradually transform into solid. The mass of liquid $\Delta \;m_{\ell }^{\left(1\right)}$ in close contact with $\xi_{1}\left(t\right)$ that is transformed into solid phase between *t* and t+Δt can be obtained through a mass balance of the solid phase in region 1 as follows:(9)$$\Delta \;m_{\ell }^{\left(1\right)}=\rho_{s}\;\left(L\left(t+\Delta \;t\right)-xi_{1}\left(t+\Delta \;t\right)\right)-\rho_{s}\;\left(L\left(t\right)-xi_{1}\left(t\right)\right).$$
Here, the first term on the right hand side represents the mass of solid in region 1 at t+Δt and the second term is the mass of solid at time *t*. Mass balance implies that $\Delta \;m_{\ell }^{\left(1\right)}$ is equivalent to the mass of solid in region 1 that is formed between *t* and t+Δt. The last equation can be simplified as follows:(10)$$\Delta \;m_{\ell }^{\left(1\right)}=\rho_{s}\left(\Delta \;L\left(t\right)-\Delta \;\xi_{1}\left(t\right)\right),$$
where ΔL(t)=L(t+Δt)−L(t) and $\Delta \;\xi_{1}\left(t\right)=xi_{1}\left(t+\Delta \;t\right)-xi_{1}\left(t\right)$ represent the displacement of the layer thickness and interface motion between *t* and t+Δt, respectively. The solidification rate of liquid mass $\Delta \;m_{\ell }^{\left(1\right)}$ in contact with the interface at $x=\xi_{1}\left(t\right)$ is produced when $\Delta \;m_{\ell }^{\left(1\right)}$ releases thermal energy in form of latent heat during a small time interval Δt. Consequently the thermal flux $dQ_{s}^{\left(1\right)}/dt$ released at the interface position $\xi_{1}\left(t\right)$ is higher than the thermal flux absorbed from the liquid layer $dQ_{\ell }^{\left(2\right)}/dt$ at $x=\xi_{1}\left(t\right)$, as shown in Figure 2. The energy–mass balance at $x=\xi_{1}\left(t\right)$ is then, given by
(11)$$\rho_{s}\;L_{f}\left(\frac{dL(t)}{dt}-\frac{d\xi_{1}\left(t\right)}{dt}\right)=\frac{dQ_{s}^{\left(1\right)}}{dt}-\frac{dQ_{\ell }^{\left(2\right)}}{dt}.$$

The temperature profile in region 1 must be a decreasing function of *x* and must have a negative concavity at region 2 within the liquid layer, as illustrated in Figure 2. Consequently, the net thermal flux at $x=\xi_{1}\left(t\right)$ can be obtained as follows
(12)$$\frac{dQ_{s}^{\left(1\right)}}{dt}-\frac{dQ_{\ell }^{\left(2\right)}}{dt}=-k_{s}\;\frac{\partial \;T_{1}^{\left(s\right)}\left(x,t\right)}{\partial x}{|}_{x=\xi_{1}\left(t\right)}+k_{\ell }\;\frac{\partial \;T_{2}^{\left(\ell \right)}\left(x,t\right)}{\partial x}{|}_{x=\xi_{1}\left(t\right)}.$$

The rate of energy released by the liquid mass $\Delta \;m_{\ell }^{\left(1\right)}$ is equal to the net thermal flux at $\xi_{1}\left(t\right)$, and the energy–mass balance at $\xi_{1}\left(t\right)$ is given by
(13)$$\rho_{s}\;L_{f}\left(\frac{dL(t)}{dt}-\frac{d\xi_{1}\left(t\right)}{dt}\right)=-k_{s}\;\frac{\partial \;T_{1}^{\left(s\right)}\left(x,t\right)}{\partial x}{|}_{x=\xi_{1}\left(t\right)}+k_{\ell }\;\frac{\partial \;T_{2}^{\left(\ell \right)}\left(x,t\right)}{\partial x}{|}_{x=\xi_{1}\left(t\right)}$$

Additionally, some of the liquid mass $\Delta \;m_{\ell }^{\left(2\right)}$ adjacent to the solid phase at region 3, will transform into solid phase between *t* and t+Δt. The mass of solid at region 3 will increase during this time interval, since the temperature on the exterior surface is below $T_{m}$ as illustrated in Figure 2. According to mass balance, $\Delta \;m_{\ell }^{\left(2\right)}$ should be equal to the mass of solid that forms during this time interval, and is given by
(14)$$\Delta \;m_{\ell }^{\left(2\right)}=\rho_{s}\;\left(\xi_{2}\left(t+\Delta \;t\right)-\xi_{2}\left(t\right)\right).$$

The energy released by this mass of liquid as latent heat, results from the net thermal flux at $x=\xi_{2}\left(t\right)$. The mass of liquid $\Delta \;m_{\ell }^{\left(2\right)}$ releases thermal energy to the solid phase at region 3 and absorbs thermal energy from the liquid layer. The rate of energy released $dQ_{s}^{\left(3\right)}/dt$ at $\xi_{2}\left(t\right)$, exceeds the rate of energy absorbed from the liquid layer $dQ_{\ell }^{\left(2\right)}/dt$ close to the interface $\xi_{2}\left(t\right)$ when solidification takes place. Additionally, the temperature profile at region 3 is an increasing function of *x* since the temperature on the exterior surface is below the melting temperature of the PCM, as shown in Figure 2. The temperature distribution in the liquid layer close to $x=\xi_{2}\left(t\right)$ shows a positive slope due to its concavity as illustrated in Figure 2. Consequently, the net amount of thermal flux at $x=\xi_{2}\left(t\right)$ is given by:(15)$$\frac{dQ_{s}^{\left(3\right)}}{dt}-\frac{dQ_{\ell }^{\left(2\right)}}{dt}=k_{s}\;\frac{\partial \;T_{3}^{\left(s\right)}\left(x,t\right)}{\partial x}{|}_{x=\xi_{2}\left(t\right)}-k_{\ell }\;\frac{\partial \;T_{2}^{\left(\ell \right)}\left(x,t\right)}{\partial x}{|}_{x=\xi_{2}\left(t\right)}.$$

The rate of thermal energy transfer at $x=\xi_{2}\left(t\right)$ is equal to the rate of latent heat energy released by $\Delta \;m_{\ell }^{\left(2\right)}$. Applying a thermal balance between the last two equations, the local energy–mass balance at $\xi_{2}\left(t\right)$ can be obtained as follows
(16)$$\rho_{s}\;L_{f}\frac{d\xi_{2}\left(t\right)}{dt}=k_{s}\;\frac{\partial \;T_{3}^{\left(s\right)}\left(x,t\right)}{\partial x}{|}_{x=\xi_{2}\left(t\right)}-k_{\ell }\;\frac{\partial \;T_{2}^{\left(\ell \right)}\left(x,t\right)}{\partial x}{|}_{x=\xi_{2}\left(t\right)}.$$

The density that appears in Equations (Equation 13) and (Equation 16) must be the density of the phase that incorporates the correct mass balance. Conservation of the PCM’s total mass may now be applied to obtain the equation of motion for the thickness of the PCM layer. In the presence of two fronts, as shown in Figure 2, the total mass of the PCM is given by
(17)$$m\left(t\right)=\rho_{s}\;\xi_{2}\left(t\right)+\rho_{\ell }\;\left(\xi_{1}\left(t\right)-\xi_{2}\left(t\right)\right)+\rho_{s}\;\left(L\left(t\right)-\xi_{1}\left(t\right)\right).$$

The time derivative of m(t) is equal to zero when the mass of the PCM layer is conserved, and the equation of motion for L(t) is given by
(18)$$\rho_{s}\;\frac{d\xi_{2}\left(t\right)}{dt}+\rho_{\ell }\;\left(\frac{d\xi_{1}\left(t\right)}{dt}-\frac{d\xi_{2}\left(t\right)}{dt}\right)+\rho_{s}\;\left(\frac{d\;L(t)}{dt}-\frac{d\xi_{1}\left(t\right)}{dt}\right)=0.$$

Equations (Equation 13), (Equation 16), and (Equation 18) represent the equations of motion for $\xi_{1}\left(t\right)$, $\xi_{2}\left(t\right)$ and L(t) that incorporate the volume changes induced by the presence of two moving fronts during the cooling stage of the cycle (when the exterior surface is below $T_{m}$).

The liquid mass $\Delta \;m_{\ell }^{\left(1\right)}$ used to obtain Equation (Equation 13), can also be estimated by subtracting the mass of liquid $\Delta \;m_{\ell }^{\left(2\right)}$ close to $\xi_{2}\left(t\right)$ from the total mass of liquid $\Delta \;m_{\ell }$ that will transform into solid between *t* and t+Δt. The mass of liquid that will change into its solid state is given by:(19)$$m_{\ell }\left(t\right)-m_{\ell }\left(t+\Delta \;t\right)=\rho_{\ell }\left(\xi_{1}\left(t\right)-\xi_{2}\left(t\right)\right)-\rho_{\ell }\left(\xi_{1}\left(t+\Delta \;t\right)-\xi_{2}\left(t+\Delta \;t\right)\right);$$
therefore, $\Delta \;m_{\ell }^{\left(1\right)}$ can be obtained from Equation (Equation 14) and the last equation, as follows
(20)$$\Delta \;m_{\ell }^{\left(1\right)}=\rho_{\ell }\left(\xi_{1}\left(t\right)-\xi_{2}\left(t\right)\right)-\rho_{\ell }\left(\xi_{1}\left(t+\Delta \;t\right)-\xi_{2}\left(t+\Delta \;t\right)\right)-\rho_{s}\;\left(\xi_{2}\left(t+\Delta \;t\right)-\xi_{2}\left(t\right)\right).$$

The rate of transformed mass $\Delta \;m_{\ell }^{\left(1\right)}/\Delta \;t$ when Δt→0 is then:(21)$$\frac{d\;m_{\ell }^{\left(1\right)}}{dt}=\rho_{\ell }\left(\frac{d\xi_{2}\left(t\right)}{dt}-\frac{d\xi_{1}\left(t\right)}{dt}\right)-\rho_{s}\;\frac{d\xi_{2}\left(t\right)}{dt},$$
which also results by solving $\rho_{s}\;\left(d\;L\left(t\right)/dt-d\xi_{1}\left(t\right)/dt\right)$ from Equation (Equation 18). Consequently, through a mass balance at the liquid layer or from total mass conservation, an equivalent equation of motion for $\xi_{1}\left(t\right)$ is obtained as follows
(22)$$\rho_{\ell }\;L_{f}\left(\left(1-\rho_{s}/\rho_{\ell }\right)\;\frac{d\xi_{2}\left(t\right)}{dt}-\frac{d\xi_{1}\left(t\right)}{dt}\right)=-k_{s}\;\frac{\partial \;T_{1}^{\left(s\right)}\left(x,t\right)}{\partial x}{|}_{x=\xi_{1}\left(t\right)}+k_{\ell }\;\frac{\partial \;T_{2}^{\left(\ell \right)}\left(x,t\right)}{\partial x}{|}_{x=\xi_{1}\left(t\right)}.$$

Equation (Equation 22) is completely equivalent to Equation (Equation 13) due to the mass balance at the liquid layer and total mass conservation. The two-front dynamics problem is described through Equations (Equation 13), (Equation 16), and (Equation 18) or equivalently through Equations (Equation 16), (Equation 18), and (Equation 22). The net thermal flux that is shown on the right hand side of Equations (Equation 13), (Equation 16), and (Equation 22) can be obtained through the temperature field at each phase, which is found by solving the local energy balance shown in Equation (Equation 4).

### 2.3. Volume Adjustments on Front Formation and Annihilation

The problem of several front formation takes place when the temperature on the exterior surface oscillates about the melting temperature of the PCM. Three possible scenarios will take place due to the thermodynamic properties, thickness of the PCM layer and the ambient temperature oscillations considered in this work. Solid phase will form when the ambient temperature reaches $T_{m}$, decreasing towards its daily minimum, a liquid phase will form when the ambient temperature reaches $T_{m}$ and increases to its daily maximum, and the two fronts, $\xi_{1}\left(t\right)$ and $\xi_{2}\left(t\right)$ will collide during the time intervals when the ambient temperature is below the melting temperature of the PCM. The thickness of the PCM layer used in this work is such that $\xi_{1}\left(t\right)$ and $\xi_{2}\left(t\right)$ will meet at some instant when the temperature on the exterior surface is still below $T_{m}$.

In this work, volume adjustments will be introduced during the creation of one-front and during the annihilation of $\xi_{1}\left(t\right)$ and $\xi_{2}\left(t\right)$. Volume displacements of the PCM layer will be incorporated so that mass is not created or destroyed during the creation of a new phase. In this work, the supercooling of liquid is not considered, and it is assumed that the liquid–solid saturation temperature is equal to its value at thermodynamic equilibrium $T_{m}$. A new solid phase will be formed and will be thermodynamically stable when the rate of energy released to the exterior $dQ_{s}^{\left(3\right)}/dt$ by the newly formed solid phase of unknown thickness $\xi_{2}\left(t_{a}\right)$ is equal to the energy released by the liquid phase $dQ_{\ell }^{\left(2\right)}/dt$ in contact with the new phase, as shown in Figure 3. The time instant $t_{a}$ when the new solid phase is stable, corresponds to any time value when the temperature on the exterior surface is equal or just below the melting temperature of the PCM. The thermal flux within the new solid phase and close to $x=\xi_{2}\left(t_{a}\right)$ is small enough and equal to the thermal flux in the liquid layer close to $\xi_{2}\left(t_{a}\right)$, so that the new solid phase is thermodynamically stable at temperature values close to $T_{m}$. Then, the thickness of the new phase can be found as follows
(23)$$k_{s}\;\frac{\partial \;T_{3}^{\left(s\right)}\left(x,t_{a}\right)}{\partial \;x}{|}_{x=\xi_{2}\left(t_{a}\right)}=k_{\ell }\;\frac{\partial \;T_{2}^{\left(\ell \right)}\left(x,t_{a}\right)}{\partial \;x}{|}_{x=\xi_{2}\left(t_{a}\right)}.$$

The left hand side of the last equation represents the rate of energy released to the environment. We assume a linear temperature distribution within the new solid phase, and the last equation is reduced to the following:(24)$$k_{s}\;\frac{T_{m}-\left(T_{0}+\delta \;\sin\left(\omega \;t_{a}+\phi \right)\right)}{\xi_{2}\left(t_{a}\right)}=k_{\ell }\;\frac{\partial \;T_{2}^{\left(\ell \right)}\left(x,t_{a}\right)}{\partial \;x}{|}_{x=\xi_{2}\left(t_{a}\right)},$$
where the temperature on the exterior surface is $T_{0}+\delta \;\sin\left(\omega \;t_{a}+\phi \right)$ as shown in Figure 3b. The volume of the PCM layer changes due to the formation of this phase, and the thickness of the PCM layer changes to avoid mass creation during the formation of the solid phase. The temperature at x=0, when $t=t_{a}-\Delta \;t$ is just above $T_{m}$, and the total mass of the PCM layer is given by
(25)$$m\left(t_{a}-\Delta \;t\right)=\rho_{\ell }\;\xi_{1}\left(t_{a}-\Delta \;t\right)+\rho_{s}\;\left(L\left(t_{a}-\Delta \;t\right)-\xi_{1}\left(t_{a}-\Delta \;t\right)\right).$$

The total mass of the system $m(t_{a}-\Delta \;t)$ at $t=t_{a}-\Delta \;t$ should be equal to $m(t_{a})$; therefore, imposing total mass conservation during the creation of the new solid phase, an equation for the thickness of the PCM layer is obtained as follows:(26)$$\rho_{\ell }\;\xi_{1}\left(t_{a}-\Delta \;t\right)+\rho_{s}\;\left(L\left(t_{a}-\Delta \;t\right)-\xi_{1}\left(t_{a}-\Delta \;t\right)\right)=\rho_{s}\;\xi_{2}\left(t_{a}\right)+\rho_{\ell }\;\left(\xi_{1}\left(t_{a}\right)-\xi_{2}\left(t_{a}\right)\right)+\rho_{s}\;\left(L\left(t_{a}\right)-\xi_{1}\left(t_{a}\right)\right)$$
where the interface at $x=\xi_{1}\left(t_{a}\right)=\xi \left(t_{a}-\Delta \;t\right)-\left(L\left(t_{a}-\Delta \;t\right)-L\left(t_{a}\right)\right)$ is shifted to the left due to the volume shrinkage of the system during the formation of the solid phase as illustrated in Figure 3b. Substituting $x=\xi_{1}\left(t_{a}\right)=\xi \left(t_{a}-\Delta \;t\right)-\left(L\left(t_{a}-\Delta \;t\right)-L\left(t_{a}\right)\right)$ in the last equation, total mass conservation of the system between $t_{a}-\Delta \;t$ and $t_{a}$ is reduced to the following
(27)$$\rho_{s}\;\xi_{2}\left(t_{a}\right)\left(1-\rho_{\ell }/\rho_{s}\right)-\rho_{\ell }\;\left(L\left(t_{a}-\Delta \;t\right)-L\left(t_{a}\right)\right)=0.$$

Finally, Equations (Equation 24) and (Equation 27) can be solved for the thickness of the new phase $\xi_{2}\left(t_{a}\right)$ and PCM layer $L(t_{a})$.

Additionally, a thin layer of liquid phase will form when the ambient temperature reaches $T_{m}$, and increases towards the daily maximum temperature $T_{0}+\delta$. During the formation of the liquid layer, volume displacements are considered to avoid loss of total mass. The process is illustrated in Figure 4a,b, where at some time $t=t_{a}-\Delta \;t$, the temperature on the exterior surface is just below $T_{m}$ and a thin layer of solid will transform into liquid phase when the temperature shifts to some value just above $T_{m}$ at some time $t_{a}$. We do not consider overheating of solid phase during the transformation, and assume that a new liquid phase of unknown thickness $\xi_{1}\left(t_{a}\right)$ is at thermodynamic equilibrium with the adjacent solid close to the new interface. The rate of energy absorbed by the newly formed liquid layer $dQ_{\ell }^{2}/dt$ is equal to the rate of energy released to the solid phase $dQ_{s}^{\left(1\right)}/dt$ at $x=\xi_{1}\left(t_{a}\right)$, as follows
(28)$$-k_{\ell }\;\frac{\partial \;T_{2}^{\left(\ell \right)}\left(x,t_{a}\right)}{\partial \;x}{|}_{x=\xi_{1}\left(t_{a}\right)}=-k_{s}\;\frac{\partial \;T_{1}^{\left(s\right)}\left(x,t_{a}\right)}{\partial \;x}{|}_{x=\xi_{1}\left(t_{a}\right)}.$$

The temperature field within the new liquid phase is close to the melting temperature of the PCM, and a linear profile is assumed; then, last equation is simplified as follows
(29)$$k_{\ell }\frac{T_{0}+\delta \;\sin\left(\omega \;t_{a}+\phi \right)-T_{m}}{\xi_{1}\left(t_{a}\right)}=-k_{s}\;\frac{\partial \;T_{1}^{\left(s\right)}\left(x,t_{a}\right)}{\partial \;x}{|}_{x=\xi_{1}\left(t_{a}\right)}.$$

Approximating the temperature distribution in the solid phase (region 1) to a linear function at $t=t_{a}$, as illustrated in Figure 4b, the last equation can be reduced as follows:(30)$$k_{\ell }\frac{T_{0}+\delta \;\sin\left(\omega \;t_{a}+\phi \right)-T_{m}}{\xi_{1}\left(t_{a}\right)}=k_{s}\frac{T_{m}-T_{C}}{L\left(t_{a}\right)-\xi_{1}\left(t_{a}\right)},$$
where $T_{C}$ is the temperature at the interior surface $x=L(t_{a})$. The corresponding expansion of the system produced by the formation of the new liquid phase can be obtained through total mass conservation. The expansion of the PCM layer is then, given by
(31)$$\rho_{s}\;L\left(t_{a}-\Delta \;t\right)=\rho_{\ell }\;\xi_{1}\left(t_{a}\right)+\rho_{s}\;\left(L\left(t_{a}\right)-\xi_{1}\left(t_{a}\right)\right).$$

Solving the last two equations for the thickness of the new liquid phase $\xi_{1}\left(t_{a}\right)$ and the thickness of the PCM layer $L(t_{a})$, the following approximate expressions are obtained:(32)$$\xi_{1}\left(t_{a}\right)=\frac{\rho_{s}\;k_{\ell }\left(T_{\mathrm{amb}}\left(t_{a}\right)-T_{m}\right)}{\rho_{\ell }\;k_{\ell }\left(T_{\mathrm{amb}}\left(t_{a}\right)-T_{m}\right)+\rho_{s}\;k_{s}\left(T_{m}-T_{C}\right)}\;L\left(t_{a}-\Delta \;t\right)\;\mathrm{and}$$
(33)$$L\left(t_{a}\right)=\frac{\rho_{s}\;k_{\ell }\left(T_{\mathrm{amb}}\left(t_{a}\right)-T_{m}\right)+\rho_{s}\;k_{s}\left(T_{m}-T_{C}\right)}{\rho_{\ell }\;k_{\ell }\left(T_{\mathrm{amb}}\left(t_{a}\right)-T_{m}\right)+\rho_{s}\;k_{s}\left(T_{m}-T_{C}\right)}\;L\left(t_{a}-\Delta \;t\right),$$
where $T_{\mathrm{amb}}\left(t_{a}\right)=T_{0}+\delta \;\sin\left(\omega \;t_{a}+\phi \right)$ is the temperature on the exterior surface x=0 as shown in Figure 4b.

Finally, the collision between $\xi_{1}$ and $\xi_{2}$ will take place at some time $t_{a}$ during the daily lows, when the temperature on the exterior surface is below $T_{m}$ as illustrated in Figure 5. The two fronts will meet at some time $t=t_{a}$, when a thin layer of liquid with thickness $\Delta \xi \left(t_{a}\right)\ll \;L\left(t_{a}\right)$ remains as saturated liquid. The temperature distribution in the liquid layer of thickness $\Delta \xi \left(t_{a}\right)=\xi_{1}\left(t_{a}\right)-\xi_{2}\left(t_{a}\right)$ is practically equal to $T_{m}$ during the phase transition. The saturated liquid is assumed to be at thermodynamic equilibrium when the phase change starts. The transformation takes place at some time $t_{a}$, when the thermal energy released by the liquid layer is exactly equal to the amount of latent heat that must be released to transform the mass of the remaining liquid. The total amount of thermal energy released by the liquid layer is given by
(34)$$\rho_{\ell }L_{f}\;\Delta \xi \left(t_{a}\right)=k_{s}\;\frac{\partial \;T_{3}^{\left(s\right)}\left(x,t_{a}\right)}{\partial \;x}{|}_{x=\xi_{2}\left(t_{a}\right)}-k_{s}\;\frac{\partial \;T_{1}^{\left(s\right)}\left(x,t_{a}\right)}{\partial \;x}{|}_{x=\xi_{1}\left(t_{a}\right)}.$$

The thickness of liquid layer $\Delta \xi (t_{a})$, when the phase change takes place can be found from the last equation as follows
(35)$$\Delta \xi \left(t_{a}\right)=\frac{k_{s}}{\rho_{\ell }L_{f}}\left(\frac{\partial \;T_{3}^{\left(s\right)}\left(x,t_{a}\right)}{\partial \;x}{|}_{x=\xi_{2}\left(t_{a}\right)}-\frac{\partial \;T_{1}^{\left(s\right)}\left(x,t_{a}\right)}{\partial \;x}{|}_{x=\xi_{1}\left(t_{a}\right)}\right)$$

The thickness of the PCM layer must decrease to avoid mass creation during the transformation to its solid state. Mass conservation can be used to find the thickness of the system once the liquid layer is transformed into its solid form as follows:(36)$$L^{{}^{\prime }}\left(t_{a}\right)=L\left(t_{a}\right)-\Delta \xi \left(t_{a}\right)\left(1-\rho_{\ell }/\rho_{s}\right),$$
where $L^{{}^{\prime }}\left(t_{a}\right)$ represents the thickness of the system, once the liquid layer changes to its solid state.

## 3. Thermal Energy Released (Absorbed): Transient and Steady Periodic Regimes

The energy released (absorbed) by the PCM layer, when the exterior surface is subjected to temperature oscillations about $T_{m}$ will be discussed. First, we describe the energy released during the formation of a solid front $\xi_{2}$ and in the presence of two liquid–solid interfaces. Later, the energy released during the collision of two fronts and the energy released (absorbed) by the PCM layer in its solid state will be described. Finally, the energy absorbed during the formation of a liquid phase and the thermal energy absorbed (released) in the presence of a one interface will be discussed.

In this work, the sensible and latent heat released (absorbed) by the PCM layer are obtained independently. The thermal energy released (absorbed) by the system is estimated as the sum of the sensible and latent heat. The sensible heat released (absorbed) corresponds to internal energy differences in the PCM, produced by temperature changes from an initial state at $t=t_{a}$ to a final state at $t=t_{b}$. Sensible heat estimations are performed through the integral of the entire temperature profiles in the PCM layer and provides complete information on the temperature distribution within the system. The FEM will be used to solve the proposed model and will be compared with the solutions estimated with the HBIM. The sensible heat released (absorbed) will be used as an indirect comparison between the temperature fields according to each method, and verify the consistency of the numerical and semi-analytical solutions. Additionally, it is observed that most of the contributions to the thermal oscillations come from the latent heat released (absorbed) by the PCM layer. The authors of Ref. [25] estimate the thermal energy released(absorbed) by the PCM layer through the time integral of the thermal flux that enters and exits the system. The net thermal flux through the layer only provides information on the behavior of the spatial derivative of the temperature at the exterior and interior surface.

### 3.1. Thermal Energy Released: Two-Front Configuration

The three phase configuration with two liquid–solid interfaces is present when the exterior temperature is below $T_{m}$. A thin solid slab will form when the ambient temperature reaches $T_{m}$ and evolves towards the daily lowest temperature values. The system releases thermal energy as latent heat during the formation of a thin solid layer of thickness $\xi_{2}\left(t_{a}\right)$. The formation of the solid layer takes place at some time $t_{a}$ when the ambient temperature is just below $T_{m}$ (supercooling of liquid is not considered). The thickness of the thin solid layer $\xi_{2}\left(t_{a}\right)$ can be found by solving Equations (Equation 24) and (Equation 27). Consequently, the latent heat released during the formation of this solid layer is given by
(37)$$\Delta \;Q_{f}=\rho_{s}\;L_{f}\xi_{2}\left(t_{a}\right)$$

Figure 6, shows the two-front configuration of the PCM layer at two different time instants $t_{a}$ and $t_{b}$. The sensible heat released between $t_{a}$ and $t_{b}$ can be conceived through differences between the internal energy of the solid, the internal energy of the liquid mass that will be transformed into solid phase and the internal energy changes of the liquid that will not transform during this time interval $\Delta \;t=t_{b}-t_{a}$.

In Figure 6, $\Delta \;\xi_{1}^{\left(s\right)}=L\left(t_{a}\right)-\xi_{1}\left(t_{a}\right)$ represents the thickness of the solid phase in region 1 at time $t=t_{a}$. After the phase transition, a fraction of the mass in the liquid layer between $\xi_{2}\left(t_{a}\right)$ and $\xi_{1}\left(t_{a}\right)$ will transform to its solid phase. The interior surface will shift to the left a distance equal to $\Delta \;L=L\left(t_{a}\right)-L\left(t_{b}\right)$ as shown in Figure 6. The shift represents the volume displacement produced by the transformation of liquid into solid phase at constant pressure. The original solid in region 1, will be shifted to the left a distance $\xi_{1}\left(t_{a}\right)-\xi_{1}^{{}^{\prime }}\left(t_{b}\right)$ as illustrated in Figure 6b. The shift can be found by applying mass conservation to the mass of solid in region 1, and solve for $\xi_{1}^{{}^{\prime }}\left(t_{b}\right)$ as follows
(38)$$\xi_{1}^{{}^{\prime }}\left(t_{b}\right)=\xi_{1}\left(t_{a}\right)+\left(L\left(t_{b}\right)-L\left(t_{a}\right)\right).$$

The internal energy change of the solid phase $\Delta \;U_{1}$ between $t_{a}$ and $t_{b}$ can now be obtained, and is given by:(39)$$\begin{array}{ccc} \Delta \;U_{1} & = & \rho_{s}\;C_{s}\;\int_{0}^{\xi_{2}\left(t_{a}\right)}\left(T_{3}^{\left(s\right)}\left(x,t_{b}\right)-T_{3}^{\left(s\right)}\left(x,t_{a}\right)\right)dx+\\ & & \rho_{s}\;C_{s}\;\left[\int_{\xi_{1}^{{}^{\prime }}\left(t_{b}\right)}^{L(t_{b})}T_{1}^{\left(s\right)}\left(x,t_{b}\right)dx-\int_{\xi_{1}\left(t_{a}\right)}^{L(t_{a})}T_{1}^{\left(s\right)}\left(x,t_{a}\right)dx\right], \end{array}$$
where the value of $\xi_{1}^{{}^{\prime }}\left(t_{b}\right)$ is given by Equation (Equation 38).

The internal energy change of the liquid mass that is not transformed into solid between $t_{a}$ and $t_{b}$ is given by
(40)$$\Delta \;U_{2}=\rho_{\ell }\;C_{\ell }\;\int_{\xi_{2}\left(t_{b}\right)}^{\xi_{1}\left(t_{b}\right)}\left(T_{2}^{\left(\ell \right)}\left(x,t_{b}\right)-T_{2}^{\left(\ell \right)}\left(x,t_{a}\right)\right)dx$$

The total thickness in the liquid layer that will be transformed into solid during the time interval $\Delta \;t=t_{b}-t_{a}$ is shown in Figure 6a. The temperature of the liquid mass that belongs to the regions in direct contact with $\xi_{1}\left(t_{a}\right)$ and $\xi_{2}\left(t_{a}\right)$ shown in Figure 6a will change from an initial value at $t=t_{a}$ to the melting temperature $T_{m}$, just before the phase transition at some time $t=t_{pt}$, between $t_{a}$ and $t_{b}$. The total thickness of this fraction of liquid is: $\Delta \;\xi_{\ell }=\Delta \;\xi_{2}^{\left(\ell \right)}+\Delta \;\xi_{1}^{\left(\ell \right)}$, where $\Delta \;\xi_{2}^{\left(\ell \right)}=\xi_{2}\left(t_{pt}\right)-\xi_{2}\left(t_{a}\right)$ and $\Delta \;\xi_{1}^{\left(\ell \right)}=\xi_{1}\left(t_{a}\right)-\xi_{1}\left(t_{tp}\right)$. Mass conservation can be applied to the fraction of liquid close to $\xi_{2}\left(t_{a}\right)$ as follows:(41)$$\rho_{\ell }\left(\xi_{2}\left(t_{pt}\right)-\xi_{2}\left(t_{a}\right)\right)=\rho_{s}\left(\xi_{2}\left(t_{b}\right)-\xi_{2}\left(t_{a}\right)\right),$$
where the right hand side represents the mass of this fraction of liquid in its solid state. The expression for $\xi_{2}\left(t_{pt}\right)$ can be found in terms of the known variables $\xi_{2}\left(t_{a}\right)$ and $\xi_{2}\left(t_{b}\right)$, as follows
(42)$$\xi_{2}\left(t_{pt}\right)=\xi_{2}\left(t_{a}\right)+\frac{\rho_{s}}{\rho_{\ell }}\left(\xi_{2}\left(t_{b}\right)-\xi_{2}\left(t_{a}\right)\right).$$

Additionally, mass conservation can be applied to the fraction of liquid close to $\xi_{1}\left(t_{a}\right)$ in the following manner:(43)$$\rho_{\ell }\left(\xi_{1}\left(t_{a}\right)-\xi_{1}\left(t_{pt}\right)\right)=\rho_{s}\left(\xi_{1}^{{}^{\prime }}\left(t_{b}\right)-\xi_{1}\left(t_{b}\right)\right),$$
where $\xi_{1}^{{}^{\prime }}\left(t_{b}\right)$ is given by Equation (Equation 38). Consequently, according to the last equation $\xi_{1}\left(t_{pt}\right)$ is given by
(44)$$\xi_{1}\left(t_{pt}\right)=\xi_{1}\left(t_{a}\right)-\frac{\rho_{s}}{\rho_{\ell }}\left(\xi_{1}\left(t_{a}\right)-\xi_{1}\left(t_{b}\right)+L\left(t_{b}\right)-L\left(t_{a}\right)\right).$$

The internal energy change of the liquid mass that will transform to its solid phase, from an initial state with temperature $T_{2}^{\left(\ell \right)}\left(x,t_{a}\right)$ to a state at the melting temperature of the PCM (saturated liquid), is given by:(45)$$\begin{array}{ccc} \Delta \;U_{3} & = & \rho_{\ell }\;C_{\ell }\left[\left(\xi_{2}\left(t_{pt}\right)-\xi_{2}\left(t_{a}\right)\right)\;T_{m}-\int_{\xi_{2}\left(t_{a}\right)}^{\xi_{2}\left(t_{pt}\right)}T_{2}^{\left(\ell \right)}\left(x,t_{a}\right)dx\right]+\\ & & \rho_{\ell }\;C_{\ell }\left[\left(\xi_{1}\left(t_{a}\right)-\xi_{1}\left(t_{pt}\right)\right)\;T_{m}-\int_{\xi_{1}\left(t_{pt}\right)}^{\xi_{1}\left(t_{a}\right)}T_{2}^{\left(\ell \right)}\left(x,t_{a}\right)dx\right], \end{array}$$
where $\xi_{2}\left(t_{pt}\right)$ and $\xi_{1}\left(t_{pt}\right)$ are given by Equations (Equation 42) and (Equation 44), respectively. Finally, when this fraction of liquid is transformed into solid, it will release sensible heat from an initial state at $t=t_{tp}$ as saturated solid to its final state with a temperature distribution $T_{3}^{\left(s\right)}\left(x,t_{b}\right)$ between $x=\xi_{2}\left(t_{a}\right)$ and $x=\xi_{2}\left(t_{b}\right)$, and $T_{1}^{\left(s\right)}\left(x,t_{b}\right)$ between $x=\xi_{1}\left(t_{b}\right)$ and $\xi_{1}^{{}^{\prime }}\left(t_{b}\right)$. The internal energy change of this fraction of transformed liquid and now in its solid state, is given by:(46)$$\begin{array}{ccc} \Delta \;U_{4} & = & \rho_{s}\;C_{s}\left[\int_{\xi_{1}\left(t_{b}\right)}^{\xi_{1}^{{}^{\prime }}\left(t_{b}\right)}T_{1}^{\left(s\right)}\left(x,t_{b}\right)dx-\left(\xi_{1}^{{}^{\prime }}\left(t_{b}\right)-\xi_{1}\left(t_{b}\right)\right)\;T_{m}\right]+\\ & & \rho_{s}\;C_{s}\left[\int_{\xi_{2}\left(t_{a}\right)}^{\xi_{2}\left(t_{b}\right)}T_{3}^{\left(s\right)}\left(x,t_{b}\right)dx-\left(\xi_{2}\left(t_{b}\right)-\xi_{2}\left(t_{a}\right)\right)\;T_{m}\right], \end{array}$$
where $\xi_{1}^{{}^{\prime }}\left(t_{b}\right)$ is given by Equation (Equation 38).

The latent heat released between $t_{a}$ and $t_{b}$ can be obtained by estimating the fraction of liquid that is transformed into solid. The liquid mass $\Delta \;m_{\ell }$ that experiences the phase transition may be obtained by subtracting the liquid mass at $t=t_{b}$ from the liquid mass at $t=t_{a}$, as follows:(47)$$\Delta \;m_{\ell }=\rho_{\ell }\left(\xi_{1}\left(t_{a}\right)-\xi_{2}\left(t_{a}\right)\right)-\rho_{\ell }\left(\xi_{1}\left(t_{b}\right)-\xi_{2}\left(t_{b}\right)\right);$$
therefore, the latent heat released during this time interval is given by
(48)$$\Delta \;Q_{f}=\rho_{\ell }\;L_{f}\left(\left(\xi_{1}\left(t_{b}\right)-\xi_{1}\left(t_{a}\right)\right)+\left(\xi_{2}\left(t_{a}\right)-\xi_{2}\left(t_{b}\right)\right)\right).$$

According to the initial and final positions of $\xi_{1}$ and $\xi_{2}$ shown in Figure 6a,b, the latent heat $\Delta \;Q_{f}$ given by Equation (Equation 48) will be negative.

### 3.2. Thermal Energy Released (Absorbed): Single Solid Phase

The collision of the two fronts $\xi_{1}$ and $\xi_{2}$ takes place at some time $t=t_{a}$, as described in the previous section. The collision, implies the release of thermal energy through latent heat, that results from the transformation of the remaining liquid layer of thickness $\Delta \;\xi =\xi_{1}\left(t_{a}\right)-\xi_{2}\left(t_{a}\right)$, into its solid state. The latent heat released during the transformation of a thin liquid layer of thickness Δξ is given by
(49)$$\Delta \;Q_{f}=\rho_{\ell }\;L_{f}\left(\xi_{2}\left(t_{a}\right)-\xi_{1}\left(t_{a}\right)\right).$$
Here, $\Delta \;Q_{f}$ is also negative, since liquid is transformed into solid phase. The thickness of the remaining liquid layer is: $\Delta \xi \left(t_{a}\right)\ll \;L\left(t_{a}\right)$, and we assume that the phase transition to its solid state is almost instantaneous. The thickness of this layer in its solid form can be obtained through mass conservation as follows:(50)$$\rho_{\ell }\left(\xi_{1}\left(t_{a}\right)-\xi_{2}\left(t_{a}\right)\right)=\rho_{s}\left(\xi_{1}^{{}^{\prime }}\left(t_{a}\right)-\xi_{2}\left(t_{a}\right)\right),$$
where $\xi_{1}^{{}^{\prime }}\left(t_{a}\right)$ is shown in Figure 5b and represents the position of the shifted interface after solidification of the liquid layer. The last equation can be solved to obtain an expression for $\xi_{1}^{{}^{\prime }}\left(t_{a}\right)$, which is given by
(51)$$\xi_{1}^{{}^{\prime }}\left(t_{a}\right)=\xi_{2}\left(t_{a}\right)+\frac{\rho_{\ell }}{\rho_{s}}\left(\xi_{1}\left(t_{a}\right)-\xi_{2}\left(t_{a}\right)\right).$$

The solid at region 1 and illustrated in Figure 5a is shifted by an amount $\Delta \;L=\xi_{1}\left(t_{a}\right)-\xi_{1}^{{}^{\prime }}\left(t_{a}\right)$ as shown in Figure 5b, and the thickness $L^{{}^{\prime }}\left(t_{a}\right)$ of the PCM layer after the phase change is given by Equation (Equation 36). The PCM layer only releases/absorbs thermal energy as sensible heat between $t=t_{a}$ and $t=t_{b}$ during the cooling stage of the cycle. Then, the internal energy change of the solid phase is given by:(52)$$\begin{array}{ccc} \Delta \;U_{s} & = & \rho_{s}\;C_{s}\left[\int_{0}^{\xi_{2}\left(t_{a}\right)}\left(T_{3}^{\left(s\right)}\left(x,t_{b}\right)-T_{3}^{\left(s\right)}\left(x,t_{a}\right)\right)dx+\int_{\xi_{2}\left(ta\right)}^{\xi_{1}^{{}^{\prime }}\left(t_{a}\right)}\left(T_{2}^{\left(s\right)}\left(x,t_{b}\right)-T_{2}^{\left(s\right)}\left(x,t_{a}\right)\right)dx\right]+\\ & & \rho_{s}\;C_{s}\int_{\xi_{1}^{{}^{\prime }}\left(ta\right)}^{L^{{}^{\prime }}\left(t_{a}\right)}\left(T_{1}^{\left(s\right)}\left(x,t_{b}\right)-T_{1}^{\left(s\right)}\left(x,t_{a}\right)\right)dx, \end{array}$$
where $\xi_{1}^{{}^{\prime }}\left(t_{a}\right)$ is given by Equation (Equation 51) and shown in Figure 5b. The thickness of the PCM layer in its solid state $L^{{}^{\prime }}\left(t_{a}\right)$ and the positions $x=\xi_{1}^{{}^{\prime }}\left(t_{a}\right)$ and $x=\xi_{2}\left(t_{a}\right)$ only represent coordinates or reference positions when the PCM is releasing/absorbing sensible heat during the cooling stage. The values of $\xi_{1}^{{}^{\prime }}\left(t_{a}\right)$ and $\xi_{2}\left(t_{a}\right)$ are constant in time and represent the location of the interfaces at the time of the collision and after the transition of the liquid layer, as shown in Figure 5.

### 3.3. Thermal Energy Absorbed (Released): One-Front Configuration

The two phase configuration in the presence of one liquid-solid interface is observed when the exterior temperature is above $T_{m}$. A thin liquid slab will form when the ambient temperature reaches the melting temperature $T_{m}$ of the PCM, and evolves towards the maximum temperature values of the cycle. The system absorbs latent heat during the formation of a thin liquid layer of thickness $\xi_{1}\left(t_{a}\right)$, given by Equation (Equation 32). The latent heat absorbed during the formation of the liquid layer can be obtained as follows:(53)$$\Delta \;Q_{f}=\rho_{\ell }\;L_{f}\xi_{1}\left(t_{a}\right),$$
where $t_{a}$ is the time value when the liquid layer is already formed. In the presence of one moving front, the PCM will absorb (release) energy through sensible and latent heat. The PCM layer will experience melting (solidification) with approximately the same frequency as the temperature oscillations.

#### 3.3.1. Thermal Energy Absorbed: Melting

The PCM layer will absorb thermal energy from the environment, when the liquid-solid interface $\xi_{1}\left(t\right)$ moves towards the interior surface, as illustrated in Figure 7a,b. The sensible heat absorbed by the initial liquid mass at $t=t_{a}$, during the time interval $\Delta \;t=t_{b}-t_{a}$ is given by:(54)$$\Delta \;U_{1}^{\left(m\right)}=\rho_{\ell }\;C_{\ell }\left[\int_{0}^{\xi_{1}\left(t_{a}\right)}\left(T_{2}^{\left(\ell \right)}\left(x,t_{b}\right)-T_{2}^{\left(\ell \right)}\left(x,t_{a}\right)\right)dx\right],$$
where the super index (m) in $\Delta \;U^{\left(m\right)}$ is used to specify internal energy changes during the melting stage, when $\xi_{1}\left(t_{b}\right)>\xi_{1}\left(t_{a}\right)$. Next, we must estimate the internal energy change of the solid layer, that will not be melted during the time interval $\Delta \;t=t_{b}-t_{a}$. In order to obtain $\Delta \;U_{2}^{\left(m\right)}$, mass conservation is used to determine the thickness of solid $\Delta \xi_{1}\left(t_{a}\right)=\xi_{1}^{\prime \prime }\left(t_{a}\right)-\xi_{1}\left(t_{a}\right)$ at time $t_{a}$ and shown in Figure 7a, that will change to its liquid state. Applying mass conservation, the thickness of this fraction of solid can be obtained as follows:(55)$$\rho_{s}\;\left(\xi_{1}^{\prime \prime }\left(t_{a}\right)-\xi_{1}\left(t_{a}\right)\right)=\rho_{\ell }\;\left(\xi_{1}\left(t_{b}\right)-\xi_{1}\left(t_{a}\right)\right),$$
where $\xi_{1}^{\prime \prime }\left(t_{a}\right)$ is shown in Figure 7a, and solving for $\xi_{1}^{\prime \prime }\left(t_{a}\right)$ from the last equation; then,
(56)$$\xi_{1}^{\prime \prime }\left(t_{a}\right)=\xi_{1}\left(t_{a}\right)+\frac{\rho_{\ell }}{\rho_{s}}\left(\xi_{1}\left(t_{b}\right)-\xi_{1}\left(t_{a}\right)\right).$$

The internal energy change of the unmelted mass of solid is given by:(57)$$\Delta \;U_{2}^{\left(m\right)}=\rho_{s}\;C_{s}\left[\int_{\xi_{1}\left(t_{b}\right)}^{L_{(}t_{b})}T_{1}^{\left(s\right)}\left(x,t_{b}\right)dx-\int_{\xi_{1}^{\prime \prime }\left(t_{a}\right)}^{L_{(}t_{a})}T_{1}^{\left(s\right)}\left(x,t_{a}\right)dx\right],$$
where $\xi_{1}^{\prime \prime }\left(t_{a}\right)$ is given by Equation (Equation 56).

The mass of solid that will melt during this time interval, will absorb sensible heat from an initial state at $t_{a}$, where the temperature distribution is $T_{1}^{\left(s\right)}\left(x,t_{a}\right)$ to a final state at some time $t_{pt}$ between $t_{a}$ and $t_{b}$, when the solid is at the melting temperature (saturated solid). The sensible heat absorbed by the fraction of melted solid between $t_{a}$ and $t_{pt}$ is given by
(58)$$\Delta \;U_{3}^{\left(m\right)}=\rho_{s}\;C_{s}\left(\xi_{1}^{\prime \prime }\left(t_{a}\right)-\xi_{1}\left(t_{a}\right)\right)\;T_{m}-\rho_{s}\;C_{s}\int_{\xi_{1}\left(t_{a}\right)}^{\xi_{1}^{\prime \prime }\left(t_{a}\right)}T_{1}^{\left(s\right)}\left(x,t_{a}\right)dx.$$

Finally, this mass of solid will absorb thermal energy from an initial state after the phase transition at some time between $t_{pt}$ as saturated liquid, to a final state at $t_{b}$ with a temperature distribution $T_{2}^{\left(\ell \right)}\left(x,t_{b}\right)$. Then, the sensible heat absorbed by this fraction of melted solid in its liquid phase, is given by
(59)$$\Delta \;U_{4}^{\left(m\right)}=\rho_{\ell }\;C_{\ell }\int_{\xi_{1}\left(t_{a}\right)}^{\xi_{1}\left(t_{b}\right)}T_{2}^{\left(\ell \right)}\left(x,t_{b}\right)dx-\rho_{\ell }\;C_{\ell }\left(\xi_{1}\left(t_{b}\right)-\xi_{1}\left(t_{a}\right)\right)\;T_{m}.$$

The latent heat absorbed can be easily found by using the mass of melted solid (newly formed liquid), as follows
(60)$$\Delta \;Q_{f}=\rho_{\ell }\;L_{f}\left(\xi_{1}\left(t_{b}\right)-\xi_{1}\left(t_{a}\right)\right).$$

Equations (Equation 54) and (Equation 57)–(Equation 60), can only be applied during the melting stage, and in the presence of one interface.

#### 3.3.2. Thermal Energy Released: Solidification

The solidification process presents a different scenario since the volume of the PCM layer is reduced during this part of the cycle. First, we consider the internal energy change experienced by the layer of liquid phase that will not transform into its solid phase between an initial state at $t_{a}$ and a final state at $t_{b}$. The thickness of this fraction of liquid is equal to $\xi_{1}\left(t_{b}\right)$ as illustrated in Figure 7c. The internal energy change experienced by this mass of liquid is given by:(61)$$\Delta \;U_{1}^{\left(s\right)}=\rho_{\ell }\;C_{\ell }\int_{0}^{\xi_{1}\left(t_{b}\right)}\left(T_{2}^{\left(\ell \right)}\left(x,t_{b}\right)-T_{2}^{\left(\ell \right)}\left(x,t_{a}\right)\right)dx,$$
where the superindex (s) represents the solidification stage and $\Delta \;U_{1}^{\left(s\right)}$ is the internal energy change experienced by this fraction of liquid phase. The thickness $L\left(t_{a}\right)-\xi_{1}\left(t_{a}\right)$ of the solid phase at $t=t_{a}$ and illustrated in Figure 7c, remains constant during the phase change process. This fraction of solid is shifted to the left a distance equal to $\Delta \;L=L\left(t_{a}\right)-L\left(t_{b}\right)$, due to the shrinkage of the liquid layer in contact with $\xi_{1}\left(t_{a}\right)$, as shown in Figure 7c,d. The contraction of the PCM layer ΔL can be used to determine the thickness of the liquid layer after the phase change as illustrated in Figure 7d, as follows:(62)$$\xi_{1}\left(t_{a}\right)-\xi_{1}^{\prime \prime \prime }\left(t_{b}\right)=L\left(t_{a}\right)-L\left(t_{b}\right).$$
Consequently, $\xi_{1}^{\prime \prime \prime }\left(t_{b}\right)$ is given by
(63)$$\xi_{1}^{\prime \prime \prime }\left(t_{b}\right)=\xi_{1}\left(t_{a}\right)+L\left(t_{b}\right)-L\left(t_{a}\right).$$

The internal energy change experienced by the initial mass of solid can be obtained as follows:(64)$$\Delta \;U_{2}^{\left(s\right)}=\rho_{s}\;C_{s}\left[\int_{\xi_{1}^{\prime \prime \prime }\left(t_{b}\right)}^{L(t_{b})}T_{1}^{\left(s\right)}\left(x,t_{b}\right)dx-\int_{\xi_{1}\left(t_{a}\right)}^{L(t_{a})}T_{1}^{\left(s\right)}\left(x,t_{a}\right)dx\right],$$
where $\xi_{1}^{\prime \prime \prime }\left(t_{b}\right)$ is shown in Figure 7d and given by Equation (Equation 63). The fraction of liquid that will be transformed to its solid phase, will release sensible heat by changing from an initial state at $t=t_{a}$ to a saturated liquid state at some time $t=t_{pt}$ between $t_{a}$ and $t_{b}$. The thickness of this fraction of liquid is $\xi_{1}\left(t_{a}\right)-\xi_{1}\left(t_{b}\right)$ and is illustrated in Figure 7c. The sensible heat released during this process can be obtained as follows
(65)$$\Delta \;U_{3}^{\left(s\right)}=\rho_{\ell }\;C_{\ell }\left(\xi_{1}\left(t_{a}\right)-\xi_{1}\left(t_{b}\right)\right)\;T_{m}-\rho_{\ell }\;C_{\ell }\int_{\xi_{1}\left(t_{b}\right)}^{\xi_{1}\left(t_{a}\right)}T_{2}^{\left(\ell \right)}\left(x,t_{a}\right)dx.$$
Additionally, when this mass of liquid changes to its solid phase, it will release sensible heat from an initial state as saturated solid at $t=t_{pt}$, to a final state at $t=t_{b}$, where its temperature distribution is $T_{1}^{\left(s\right)}\left(x,t_{b}\right)$. The sensible heat released by the newly formed solid phase is given by:(66)$$\Delta \;U_{4}^{\left(s\right)}=\rho_{s}\;C_{s}\int_{\xi_{1}\left(t_{b}\right)}^{\xi_{1}^{\prime \prime \prime }\left(t_{b}\right)}T_{1}^{\left(s\right)}\left(x,t_{b}\right)dx-\rho_{s}\;C_{s}\left(\xi_{1}^{\prime \prime \prime }\left(t_{b}\right)-\xi_{1}\left(t_{b}\right)\right)\;T_{m},$$
where $\xi_{1}^{\prime \prime \prime }\left(t_{b}\right)$ is given by Equation (Equation 63).

Finally, the latent heat released during the phase transition is given by
(67)$$\Delta \;Q_{f}=\rho_{\ell }\;L_{f}\left(\xi_{1}\left(t_{b}\right)-\xi_{1}\left(t_{a}\right)\right)$$

## 4. Numerical and Semi-Analytical Methods

Front tracking methods are applied to solve the model described in Section 2 with the volume corrections proposed in this work. The HBIM is used to find approximate semi-analytical solutions, and a FEM with first order Lagrange interpolating functions is used to verify the consistency of the semi-analytical solutions. The HBIM demands continuity and smoothness of the temperature profile in each phase. The isothermal boundary condition at the liquid-solid front, introduces a discontinuity in the spacial derivative of the temperature when two fronts collide and the entire PCM layer is in its solid phase. In this work, the HBIM is modified by introducing a local energy balance at the collision site and the solutions are verified through comparison with the FEM.

### 4.1. Heat Balance Integral Method

The HBIM is adapted to find approximate analytical solutions to the model described in Section 2. Continuous and smooth temperature profiles at each phase are required to apply the HBIM used in Refs. [14,30,31]. The HBIM consists on proposing a polynomial function for the temperature at each phase. The space integral of Equation (Equation 4) is performed at each phase, to obtain a set of ordinary differential equations (ODE) in time. Quadratic functions of *x* that satisfy the boundary conditions, are proposed in this work. The following temperature profiles are proposed to solve the two-front configuration scenario illustrated in Figure 2, as follows:(68)$$\begin{array}{ccc} T_{1}^{\left(s\right)}\left(x,t\right) & = & a_{1}\left(t\right)\left(x-\xi_{1}\left(t\right)\right)+b_{1}\left(t\right){\left(x-\xi_{1}\left(t\right)\right)}^{2}+T_{m},\;\;\;\;\mathrm{for}\;\;\;\;\xi_{1}\left(t\right)\leq \;x\leq \;L\left(t\right),\\ T_{2}^{\left(\ell \right)}\left(x,t\right) & = & a_{2}\left(t\right)\left(x-\xi_{1}\left(t\right)\right)+b_{2}\left(t\right){\left(x-\xi_{1}\left(t\right)\right)}^{2}+T_{m},\;\;\;\;\mathrm{for}\;\;\;\;\xi_{2}\left(t\right)\leq \;x\leq \;\xi_{1}\left(t\right),\\ T_{3}^{\left(s\right)}\left(x,t\right) & = & a_{3}\left(t\right)\left(x-\xi_{2}\left(t\right)\right)+b_{3}\left(t\right){\left(x-\xi_{2}\left(t\right)\right)}^{2}+T_{m},\;\;\;\;\mathrm{for}\;\;\;\;0\leq \;x\leq \;\xi_{2}\left(t\right). \end{array}$$
Here, the time dependent coefficients $a_{i}\left(t\right)$ at each region i=1,2,3 can be expressed in terms of $b_{i}\left(t\right)$ through the boundary conditions given by:(69)$$\begin{array}{ccc} T_{1}^{\left(s\right)}\left(L(t),t\right) & = & T_{C},\\ T_{2}^{\left(\ell \right)}\left(\xi_{2}\left(t\right),t\right) & = & T_{m},\;\;\;\;\mathrm{and}\\ T_{3}^{\left(s\right)}\left(0,t\right) & = & T_{0}+\delta \sin\left(\omega \;t+\phi \right). \end{array}$$

The boundary condition at $x=\xi_{1}\left(t\right)$ is satisfied by the proposed temperature profiles. Applying the boundary conditions shown in the last equation the following time dependent coefficients are found:(70)$$\begin{array}{ccc} a_{1}\left(t\right) & = & b_{1}\left(t\right)\left(L\left(t\right)-\xi_{1}\left(t\right)\right)+\frac{T_{m}-T_{C}}{\xi_{1}\left(t\right)-L\left(t\right)},\\ a_{2}\left(t\right) & = & b_{2}\left(t\right)\left(\xi_{1}\left(t\right)-\xi_{2}\left(t\right)\right),\;\;\;\;\mathrm{and}\\ a_{3}\left(t\right) & = & b_{3}\left(t\right)\xi_{2}\left(t\right)+\frac{T_{m}-T_{0}-\delta \sin\left(\omega \;t+\phi \right)}{\xi_{2}\left(t\right)} \end{array}$$

The temperature profiles are substituted into Equation (Equation 4) and the result is integrated over the domain of each phase to obtain a set of three ODEs in time for the coefficients $b_{i}\left(t\right)$. The two-front configuration of the PCM layer becomes a dynamic problem for the time dependent variables $\xi_{1}\left(t\right)$, $\xi_{2}\left(t\right)$, L(t) and $b_{i}\left(t\right)$ with i=1,2,3. The three ODEs in time for the coefficients $b_{i}\left(t\right)$, along with Equations (Equation 13), (Equation 16) and (Equation 18), constitute a set of six ODEs that is solved through an explicit finite difference method with a forward first order approximation to the time derivatives.

The one-front configuration problem illustrated in Figure 1 is solved similarly. The space integral of Equation (Equation 4) is performed at each region shown in Figure 1. Two ODEs in time for the coefficients $b_{i}\left(t\right)$ with i=1,2 is obtained through the spacial integrals. The PCM layer in the presence of one-front, becomes a dynamic problem for the time dependent variables $\xi_{1}\left(t\right)$, L(t) and $b_{i}\left(t\right)$ with i=1,2 [14].

The HBIM just described, requires continuity and smoothness of each temperature profile within its domain. The method must be slightly adapted to scenarios where the temperature distribution is not smooth as illustrated in Figure 5b. The situation depicted in Figure 5b results from the collision of two fronts and just after the saturated liquid is transformed into its solid phase. The two-front collision scenario shown in Figure 5b takes place at some time $t_{a}$, when the temperature is approaching the daily minimum, and the entire PCM layer is in its solid state. The isothermal boundary condition at $\xi_{1}$ and $\xi_{2}$ is no longer required. The liquid-solid interfaces disappear and the thickness of the PCM layer becomes a constant of the motion in the absence of a phase change process. The problem consists of solving Equation (Equation 4) with the initial temperature profile illustrated in Figure 5b. A single polynomial function with the initial temperature field shown in Figure 5b is not possible, and instead a piecewise function that is consistent with the initial conditions is proposed as follows:
(71)$$\begin{array}{ccc} T_{1}^{\left(s\right)}\left(x,t\right) & = & a_{1}\left(t\right)\left(x-\xi_{1}\right)+b_{1}\left(t\right){\left(x-\xi_{1}\right)}^{2}+T_{u}\left(t\right),\;\;\;\;\mathrm{for}\;\;\;\;\xi_{1}\leq \;x\leq \;L,\\ T_{2}^{\left(s\right)}\left(x,t\right) & = & T_{u}\left(t\right),\;\;\;\;\mathrm{for}\;\;\;\;\xi_{2}\leq \;x\leq \;\xi_{1},\;\mathrm{and}\\ T_{3}^{\left(s\right)}\left(x,t\right) & = & a_{3}\left(t\right)\left(x-\xi_{2}\right)+b_{3}\left(t\right){\left(x-\xi_{2}\right)}^{2}+T_{u}\left(t\right),\;\;\;\;\mathrm{for}\;\;\;\;0\leq \;x\leq \;\xi_{2}, \end{array}$$
where $T_{u}\left(t\right)$ is the temperature at the collision site of thickness $\Delta \xi =\xi_{1}-\xi_{2}$ and initially, equal to the melting temperature of the PCM. Initially, the region of thickness Δξ represents a saturated solid. The HBIM can be applied to regions 1 and 3 through the space integral of Equation (Equation 4), as previously described. The coordinates $x=\xi_{1}$ and $x=\xi_{2}$ now represent locations in space within the solid, and define a thin solid layer of thickness Δξ≪L. In this work, instead of solving the heat equation in region 2, we assume a uniform temperature profile $T_{u}\left(t\right)$ in this thin layer that will change in time according to a local energy balance principle. Following the basic idea of energy balance, the thin solid layer of thickness Δξ, will release(absorb) thermal energy that comes from the net heat flux at $x=\xi_{1}$ and $x=\xi_{2}$, as illustrated in Figure 5b. The thin solid layer Δξ, releases thermal energy when the temperature is distributed through the PCM layer, as illustrated in Figure 5b. The rate of energy released by the solid layer Δξ during a small time interval Δt, is equal to the internal energy change $\Delta \;U_{s}$ experienced by this layer, as follows
(72)$$\frac{\Delta \;U_{s}}{\Delta \;t}=C_{s}\rho_{s}\frac{T_{u}\left(t\right)-T_{u}\left(t+\Delta \;t\right)}{\Delta \;t}.$$

The internal energy change shown through the last equation, results from the energy released to the solid in regions 1 and 3 as depicted in Figure 5b. The net rate of thermal energy transferred can be obtained as follows
(73)$$\frac{dQ_{s}^{\left(1\right)}}{dt}+\frac{dQ_{s}^{\left(3\right)}}{dt}=-k_{s}\frac{\partial \;T_{1}^{\left(s\right)}\left(x,t\right)}{\partial \;x}{|}_{x=\xi_{1}}+k_{s}\frac{\partial \;T_{3}^{\left(s\right)}\left(x,t\right)}{\partial \;x}{|}_{x=\xi_{2}}.$$

According to the last two equations, the local energy balance at region 2 in the limit Δt→0, is given by
(74)$$C_{s}\rho_{s}\frac{dT_{u}\left(t\right)}{dt}=k_{s}\frac{\partial \;T_{1}^{\left(s\right)}\left(x,t\right)}{\partial \;x}{|}_{x=\xi_{1}}-k_{s}\frac{\partial \;T_{3}^{\left(s\right)}\left(x,t\right)}{\partial \;x}{|}_{x=\xi_{2}},$$
which constitutes a differential equation for the temperature $T_{u}\left(t\right)$ in the thin solid layer between $x=\xi_{2}$ and $x=\xi_{1}$. Additionally, the energy balance given by Equation (Equation 74) may also be applied when the thin solid layer absorbs thermal energy. Finally, the time evolution of $T_{u}\left(t\right)$ is determined through the solution of Equation (Equation 74), which is used to estimate the coefficients $b_{i}\left(t\right)$ with i=1,3 through the classical HBIM, previously described.

### 4.2. Finite Element Method

The FEM was applied to find the spacial dependence of the temperature field in each phase [32]. Equation (Equation 4) was solved through the methodology described in Ref. [32], but using the linear Lagrange shape functions at each element as follows:(75)$$\begin{array}{ccc} N_{1}\left(x\right) & = & \frac{x-x_{2}}{x_{1}-x_{2}},\;\;\mathrm{and}\\ N_{2}\left(x\right) & = & \frac{x-x_{1}}{x_{2}-x_{1}}. \end{array}$$
Here $x_{1}$ and $x_{2}$ represent the nodal coordinates of any given element. The temperature field within each element is given by:(76)$$\tilde{T}\left(x,t\right)=N_{1}\left(x\right)\;{\hat{T}}_{1}\left(t\right)+N_{2}\left(x\right)\;{\hat{T}}_{2}\left(t\right),$$
where, ${\hat{T}}_{1}\left(t\right)$ and ${\hat{T}}_{2}\left(t\right)$ is the time dependent part of the temperature at each node. The time evolution of the temperature at each node was obtained through a first order approximation to the time derivative of Equation (Equation 4). The implicit finite difference scheme was applied to estimate the nodal temperatures at the next time level.

Equations (Equation 13), (Equation 16), and (Equation 18) were solved for the dynamical variables $\xi_{1}\left(t\right)$, $\xi_{2}\left(t\right)$, and L(t) by using an explicit finite difference scheme with a first order approximation to the time derivatives. The position of each interface and the thickness of the PCM layer in the next time level were used to solve the heat equation through the FEM previously mentioned [32].

## 5. Results and Discussion

The PCM used as an insulating material is octadecane. The thermal performance of a PCM layer of octadecane is determined by estimating the time evolution of the dynamical variables, the thermal energy released (absorbed) and the thermal energy released by the interior surface. The thermodynamic properties of octadecane are assumed to be constant in the temperature range considered in this work, and equal to their values close to the saturation temperature of the PCM [27]. For the liquid(solid) phase $k_{\ell }=0.152\left(k_{s}=0.334\right)\;W/m\cdot \;K$, $C_{\ell }=1.921\left(C_{s}=2.230\right)\;\mathrm{kJ}/\mathrm{kg}\cdot \;K$ and $\rho_{\ell }=776.860\left(\rho_{s}=867.914\right)\;\mathrm{kg}/m^{3}$. The liquid-solid saturation properties are: $L_{f}=236.98\;\mathrm{kJ}/\mathrm{kg}$ and $T_{m}=301.13\;K$ [27].

### 5.1. One-Front Dynamics: Transient and Steady Periodic Regimes

The one-front dynamics scenario is present when the ambient temperature oscillates above the melting temperature of Octadecane during the complete cycle. In this case, the parameters for the ambient ambient temperature are: $T_{0}=308.15\;K$, δ=5.0K and ϕ=0.73779rad. The phase angle ϕ represents a shift of the sine function, and the temperature at the inner surface is fixed at $T_{C}=295.15\;K$.

Equations (Equation 2)–(Equation 4) were solved through the FEM and the HBIM described in the previous section. Solutions were found for different values of the Stefan number defined as:(77)$$\mathrm{SteNo}=\frac{L_{f}}{C_{\ell }\left(T_{\max}-T_{m}\right)},$$
where $T_{\max}=T_{0}+\delta$ is the maximum temperature on the exterior surface in this example. Figure 8, shows the transient and steady periodic parts of the solution for the interface position $\xi_{1}\left(t\right)$ and the thickness of the PCM layer L(t). Approximate HBIM solutions are validated through FEM solutions in the one-front dynamics problem as shown in Figure 8. We have predicted lower and upper bounds with volume corrections, for the interface position and thickness of the PCM layer, as shown through Equations (Equation 5)–(Equation 8). Figure 8 shows solutions in the steady periodic regime for several values of SteNo. According to Equations (Equation 5)–(Equation 8), the interface position and thickness of the PCM layer are bounded in the steady periodic regime. The solutions are observed to oscillate within the predicted bounds for several values of SteNo.

The maxima and minima in the oscillations of $\xi_{1}\left(t\right)$ and L(t) close to the steady periodic regime, are tested by probing the solutions in the range SteNo=[008841−88.4120]. The Stefan number is modified by changing the magnitude of the latent heat. According to Equations (Equation 5)–(Equation 8), upper and lower bounds for $\xi_{1}$ and *L* close to the steady periodic regime, should not depend on $L_{f}$ when the system is subjected to non-homogeneous isothermal boundary conditions. The steady periodic solutions must approach asymptotically to the lower and upper bounds predicted through Equations (Equation 5)–(Equation 8) for low values of $L_{f}$. The asymptotic behavior close to the steady periodic regime according to Equations (Equation 5)–(Equation 8), was captured by the HBIM and the FEM solutions. Maxima and minima in the oscillations of $\xi_{1}\left(t\right)$ and L(t), were registered for several values of the Stefan number as shown in Figure 9. Melting (solidification) rates are expected to increase for low values of the latent heat as observed in Figure 9; therefore, for small Stefan numbers, the amplitude of the oscillations in the steady periodic regime is exactly bounded by the values predicted through Equations (Equation 5)–(Equation 8).

### 5.2. Two-Front Dynamics: Transient and Steady Periodic Regimes

The temperature on the exterior surface was extracted from weather data of 10 August 2021 at the city of Villahermosa, Tabasco in Mexico [28]. The temperature data was fitted to the periodic function shown in Equation (Equation 1), with the following fitting parameters: $T_{0}=302.884\;K$, δ=−5.24632K and ϕ=0.73779rad. The melting temperature of the PCM selected for the numerical and semi-analytical examples, and the ambient temperature of the selected region can give rise to the formation of several fronts. The ambient temperature oscillates around the melting temperature of octadecane, which is chosen as the PCM for the numerical and semi-analytical examples. The initial thickness of the PCM layer is $L_{0}=3.0\;\mathrm{cm}$, and a two-front configuration is observed during part of the cooling stage of the cycle. The temperature at the inner surface is $T_{C}=295.15$. The inner temperature of $T_{C}=295.15$ has been chosen to observe the two-front configuration scenario with the longest possible duration, but with inner temperatures that produce some thermal comfort. Lowest values of $T_{C}$ are possible; however, volumetric effects would be less evident.

The model described through Equations (Equation 4), (Equation 13), (Equation 16), and (Equation 18) will be solved with the FEM and HBIM discussed in the previous section. The model is equivalent to Equations (Equation 4), (Equation 16), (Equation 18), and (Equation 22), where Equation (Equation 22) results from applying total mass conservation to Equation (Equation 13). Figure 10 shows the FEM and HBIM solutions for $\xi_{1}\left(t\right)$, $\xi_{2}\left(t\right)$, and L(t) in a Octadecane layer with an initial thickness of $L_{0}=3.0\;\mathrm{cm}$. Initially, the system is almost in its liquid phase, with a thin solid layer of thickness $\xi_{2}\left(0\right)=1.0\;\mathrm{mm}$ close to the exterior surface. Additionally, a solid layer of $L\left(0\right)-\xi_{1}\left(0\right)=1.0\;\mathrm{mm}$ of thickness lies close to the interior surface. The system is close to a steady periodic regime after being exposed to the ambient temperature for 4–5 days.

Figure 10 contains the information on the front configuration during a complete cycle: two-front, one-front and single solid phase configurations. The transient part of the solution is observed for t≪1day, where the layer thickness is significantly reduced. Initially, the ambient temperature is decreasing towards the daily lows and a significant amount of thermal energy is released by the PCM layer.

Internal energy and latent heat changes were estimated during the transient and steady periodic regimes. The model described through Equations (Equation 4), (Equation 13), (Equation 16), and (Equation 18) is solved, and thermal energy changes during small time intervals of Δt=0.1s are estimated. Figure 11 shows the sensible heat, latent heat, and total energy released (absorbed) by the PCM during the transient and steady periodic regimes. The thermal energy released (absorbed) can also be found by performing the time integral of the net thermal flux through the PCM layer [25]. The net rate of thermal energy change in the PCM can be obtained through the difference between the thermal flux on the exterior and inner surface as follows
(78)$$\frac{dQ_{\mathrm{PCM}}}{dt}=\frac{dQ_{\mathrm{ext}}}{dt}-\frac{dQ_{\mathrm{in}}}{dt}.$$

The energy released (absorbed) by the PCM layer between t=0 and $t=t_{a}$ is then, given by:(79)$$\Delta \;Q_{\mathrm{PCM}}=\int_{0}^{t_{a}}\left(k_{i}\frac{\partial \;T_{j}^{\left(i\right)}\left(x,t\right)}{\partial \;x}{|}_{x=0}+k_{s}\frac{\partial \;T_{1}^{\left(s\right)}\left(x,t\right)}{\partial \;x}{|}_{x=L(t)}\right)dt,$$
where $k_{i}$ represents the thermal conductivity of phase i=ℓ,s and $T_{j}^{\left(i\right)}\left(x,t\right)$ is the temperature distribution of phase *i* within region j=2,3. The last equation may be used instead of estimating the thermal energy released (absorbed) through the process described in Section 3. On the one hand, Equation (Equation 79) contains global information related with the volumetric effects on the sensible and latent heat released (absorbed) by the PCM, and only provides information on the total energy released (absorbed). On the other hand, the process described in Section 3 includes detailed information related with the manner in which sensible and latent heats are being released or absorbed. Additionally, Equation (Equation 79) only depends on the behavior of the temperature on the exterior and inner surfaces, and does not provide information related with the entire temperature field in the PCM. The process described in Section 3 is also preferred, since the time evolution of the temperature profiles integrated over the PCM domain, can be used to perform a more consistent comparison between the numerical and semi-analytical solutions used in this work.

Thermal energy is released during the solidification stages, which are observed when the ambient temperature is below $T_{m}$ (night hours) and during the cooling part of the day. The amount of sensible heat released by the PCM in the presence of two liquid–solid interfaces is obtained by substituting the solutions at two different time values $t_{a}$ and $t_{b}$ in Equations (Equation 39), (Equation 40), (Equation 45), and (Equation 46). The sensible heat released when the system is in its pure solid state and during the intervals with lowest ambient temperatures is estimated through Equation (Equation 52). Additionally, during the solidification process, and in the presence of one-front, the sensible heat released is obtained through Equations (Equation 61) and (Equation 64)–(Equation 66). Latent heat is released during the solidification processes, during the formation of a solid layer and at the instant of collision between $\xi_{1}$ and $\xi_{2}$. The latent heat released in the presence of two fronts and one-front, was estimated through Equation (Equation 48) and (Equation 67), respectively. Latent heat released during the formation of a solid layer close to the exterior surface and at the time of collision was obtained through Equation (Equation 37) and (Equation 49), respectively. Similarly, the sensible heat absorbed during the melting process was estimated through Equations (Equation 54) and (Equation 57)–(Equation 59). Sensible heat is also absorbed when the PCM layer is in its solid state and during the intervals when the ambient temperature is increasing from its daily minimum and towards the melting temperature of the PCM. Finally, the latent heat absorbed during the formation of a liquid layer and during the melting process, is obtained through Equations (Equation 53) and (Equation 60), respectively.

Table 1 shows the highest relative percent difference (RPD) between the model discussed in this work and constant volume models commonly by other authors. The RPD was obtained for the sensible heat, latent heat and thermal energy released (absorbed) through each method used in this work. The RPD was estimated as follows:(80)$${\mathrm{RPD}}_{\Delta \;E}^{\left(\max,i\right)}=\frac{|\left(\Delta \;E_{p}-\Delta \;E_{o}\right)|}{|\Delta \;E_{p}|}\times \;100\%,$$
where $\Delta \;E_{p}$ and $\Delta \;E_{o}$ represents the sensible heat, latent heat or thermal energy released(absorbed) by the PCM layer according to the proposed model in this work and the constant volume methods used by other authors, respectively. The maximum RPD with i=1 and i=2 corresponds to the RPD between the proposed model ($\rho_{s}\neq \rho_{\ell }$) and a constant volume model with $\rho_{s}=\rho_{\ell }$, and the RPD between the proposed model ($\rho_{s}\neq \rho_{\ell }$) and a constant volume model with $\rho_{s}=\rho_{\ell }$, respectively. The maximum RPD is found when the system is close to the steady periodic regime and is shown in Table 1.

Numerical and semi-analytical solutions to models that do not incorporate volumetric effects are also shown in Figure 11. Two scenarios are considered by assuming an octadecane sample of L=3.0cm with phases of equal densities. On the one hand, the density of the liquid and solid phase are equal to $\rho_{s}=867.914\;\mathrm{kg}/m^{3}$. On the other hand, the density of both phases is equal to the liquid density $\rho_{\ell }=776.860\;\mathrm{kg}/m^{3}$. The solutions for ΔU, $\Delta Q_{f}$, ΔQ, and total mass are labeled as $\rho_{\ell }=\rho_{s}$ and $\rho_{s}=\rho_{\ell }$ in Figure 11. Volume changes expansion or shrinkage disappear by assuming a PCM with no density change during the phase transition. In absence of volume changes, total mass is implicitly conserved and an additional equation of motion for the thickness *L* of the PCM layer is not required. Thickness and total mass are constants of the motion when equal densities are assumed. Sensible and latent heats can be obtained by taking the corresponding limit of equal densities on each of the equations discussed in Section 3.

The effects of PCM expansion and shrinkage can be observed in Figure 11 and depend on the initial conditions of the problem. Initially, the system is practically in its liquid form, since $\xi_{2}\left(0\right)=1.0\;\mathrm{mm}$ and $\xi_{1}\left(0\right)=2.9\;\mathrm{cm}$. Additionally, the temperature on the exterior surface is below $T_{m}$ and evolving towards the daily minimum. The initial conditions chosen on this example produce solidification of the liquid layer with an initial thickness of $\Delta \xi_{\mathrm{liq}}\left(0\right)=\xi_{1}\left(0\right)-\xi_{2}\left(0\right)=2.8\;\mathrm{cm}$. The shrinkage of the system from its initial state to its pure solid form can be obtained through a simple mass balance where $\rho_{\ell }\Delta \xi_{\mathrm{liq}}\left(0\right)=\rho_{s}\Delta \xi_{\mathrm{sol}}$. Here, $\Delta \xi_{\mathrm{sol}}$ is the thickness of the initial mass of liquid, but in its solid form. Consequently, after solidification of the initial liquid layer, the system must shrink an amount equal to: $\Delta \xi_{\mathrm{liq}}\left(0\right)-\Delta \xi_{\mathrm{sol}}=2.938\;\mathrm{mm}$; therefore, the thickness of the PCM layer should be $L(t_{a})=2.706\;\mathrm{cm}$ at the time $t_{a}$ of the first collision, as observed in Figure 10b. Assuming for example, a PCM where the liquid phase has the same density as the PCM in its solid form $\rho_{\ell }=\rho_{s}$, the mass of the PCM layer is higher, as shown in Figure 11d. The liquid phase has a larger mass in this case, and consequently the latent heat released is significantly higher as illustrated in Figure 11b. The latent heat initially released, is shifted along the energy axis as a consequence of the extra mass that results by assuming $\rho_{\ell }=\rho_{s}$. The other scenario, when $\rho_{s}=\rho_{\ell }$, does not produce the observed shift on the latent heat, since total mass does not change significantly.

Finally, Figure 12 shows the energy released by the solid in region 1 and to the interior of the room. The energy released by the interior surface, represents the amount of thermal energy that must be removed from the room to keep a constant temperature at the inner surface of the PCM layer. The energy released by the solid phase at region 1 can be obtained through the time integral of the thermal flux on the inner surface as follows
(81)$$\Delta \;Q_{\mathrm{in}}=-k_{s}\int_{0}^{t_{a}}\frac{\partial \;T_{1}^{\left(s\right)}\left(x,t\right)}{\partial \;x}{|}_{x=L(t)}dt.$$

The energy released by the inner surface represents another way to evaluate the thermal efficiency of the PCM layer as a thermal barrier. The result is compared with the models of other authors, where volumetric effects are not considered.

The enhanced thermal performance of the PCM when equal densities are assumed, can be understood in terms of the initial state of the system and the thickness of the solid phase in region 1. On the one hand, the thickness of the PCM layer remains constant in this case, while it is reduced initially by almost 3.0mm when $\rho_{\ell }\neq \rho_{s}$, as previously discussed. Close to the steady periodic regime, the thickness of the PCM layer will oscillate between 2.7cm and 2.8cm as a consequence of this initial shrinkage. On the other hand, the thickness of the PCM layer will remain constant and equal to its initial value of L=3.0cm when equal densities are assumed. The oscillations of the interface at $x=\xi_{1}\left(t\right)$ are very similar in both cases (equal and different densities); therefore, the effective thickness of the solid phase in region 1 and close to the steady periodic regime, will be smaller when volumetric effects are incorporated into the problem. Consequently, since the temperature gradient in region 1 is $\Delta \;T=T_{m}-T_{C}$ in both cases, higher energy transfer rates at x=L(t) will be expected when $\rho_{\ell }\neq \rho_{s}$.

## 6. Conclusions

The main goal of this work consisted in estimating the effects of volume changes in the thermal performance of a PCM layer, when the external surface is exposed to temperature oscillations about the fusion point of the PCM. Despite the relatively small density variations in this material, the volumetric effects on the energy transferred by the PCM layer were significant. The initial conditions of the system produced an overall maximum volume shrinkage of 10% during the transient regime, which is observed when the two fronts collide for the first time. The latent heat, sensible heat, and thermal energy released in the transient regime are shifted along the energy axis as a result of the initial shrinkage of the PCM layer. The shift of the thermal energy initially released produces a significant relative percent difference between the model proposed in this work and the constant volume methods used by other authors. The largest relative percent difference is observed when the liquid density is equal to the solid density. The result is expected since assuming phases with densities equal to $\rho_{s}$, the total mass of the system is increased. Additionally, it is found that the system does not recover its initial thickness when the PCM layer oscillates in the steady periodic regime. The thickness of the PCM layer is significantly reduced and oscillates around a smaller value in the steady periodic regime, constituting a less effective thermal barrier as a result of the initial system shrinkage. The results indicate that the initial state of the system has a significant impact on the thermal performance of the PCM, which is overestimated when neglecting volume changes. The situation may be reversed, and thermal performance could be enhanced by using an initial state with two liquid fronts that produce a system expansion from which the PCM layer may not recover. The result will depend on the thermodynamic properties of the PCM and the temperature oscillations on the exterior surface. The volumetric effects produced by a combination of different transient states has yet to be determined. Different transient states may appear in systems with non-sinusoidal temperature oscillations. The system is unable to reach a steady periodic regime in this case, and the volumetric effects predicted in this work may be more profound.

Additional contributions to volume changes produced by considering the temperature dependence of liquid and solid densities must be addressed as well. Thermal expansion effects on the problem of several front formation will depend on the type of PCM. Ambient temperature variations may also give rise to a different phase configuration than discussed in this work when considering the thermal expansion of each phase. Finally, to the authors’ knowledge, there is little experimental evidence in the literature concerning the problem of several front formation. The present work is still limited by the experimental validations that will allow estimation of the effects predicted through the proposed model. 

## Figures and Tables

**Figure 1 molecules-27-02158-f001:**
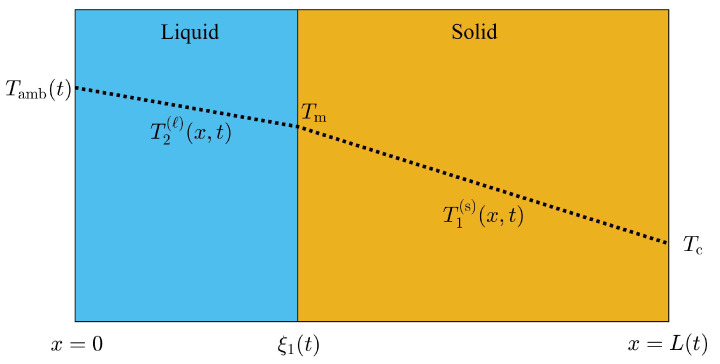
Schematic representation of the PCM layer in the presence of one liquid–solid interface or front with the temperature profiles in the liquid phase (region 2) and solid phase (region 1).

**Figure 2 molecules-27-02158-f002:**
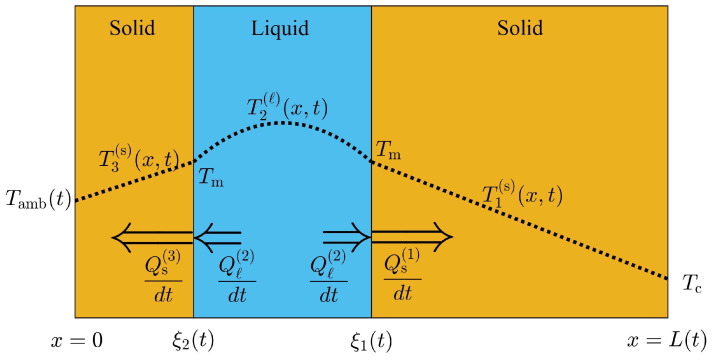
Three phase configuration with a PCM layer in the presence of two fronts. The two-front configuration is observed when the temperature on the exterior surface is below $T_{m}$. The liquid layer releases thermal energy to the solid phases at regions 1 and 3 (supercooling effects are not considered).

**Figure 3 molecules-27-02158-f003:**
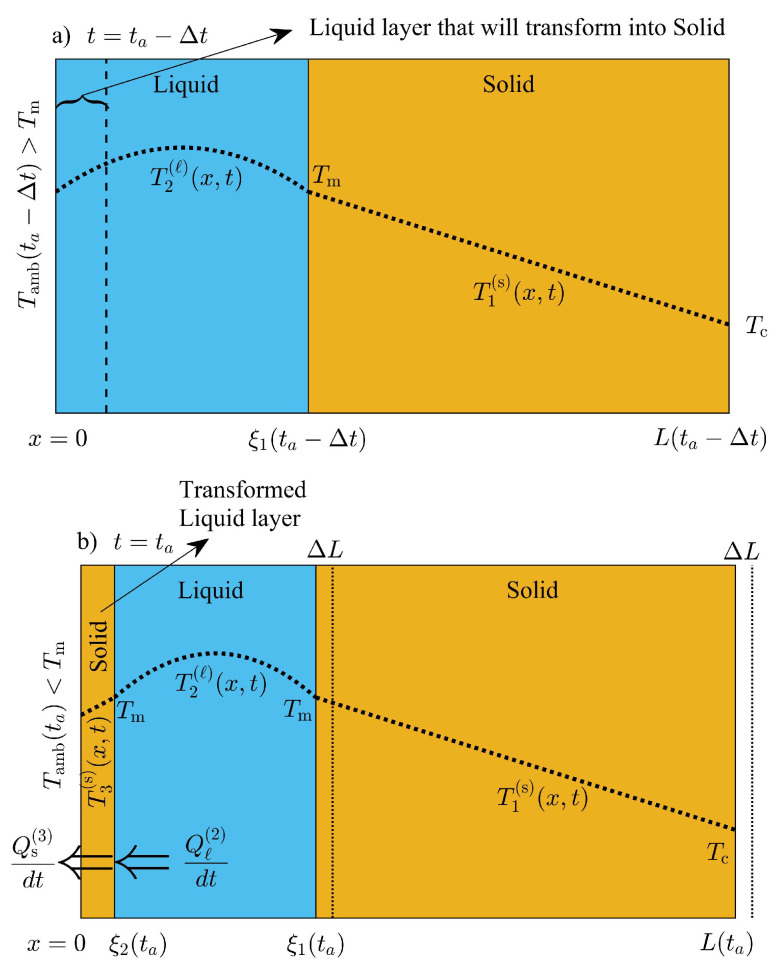
Solid phase formation close to the exterior surface produced by the change of the ambient temperature from $T_{\mathrm{amb}}\left(t_{a}-\Delta \;t\right)>T_{m}$ to $T_{\mathrm{amb}}\left(t_{a}\right)<T_{m}$ during a small time interval Δt. (**a**) Thin liquid layer that will change to its solid phase. The layer is still in its liquid form at $t=t_{a}-\Delta \;t$, when the temperature on the exterior surface is just above the melting temperature of the PCM. (**b**) Volume shrinkage $\Delta \;L=L\left(t_{a}-\Delta \;t\right)-L\left(t_{a}\right)$ produced by the shift of the interior surface, after the liquid layer in contact with the exterior surface is transformed into its solid phase.

**Figure 4 molecules-27-02158-f004:**
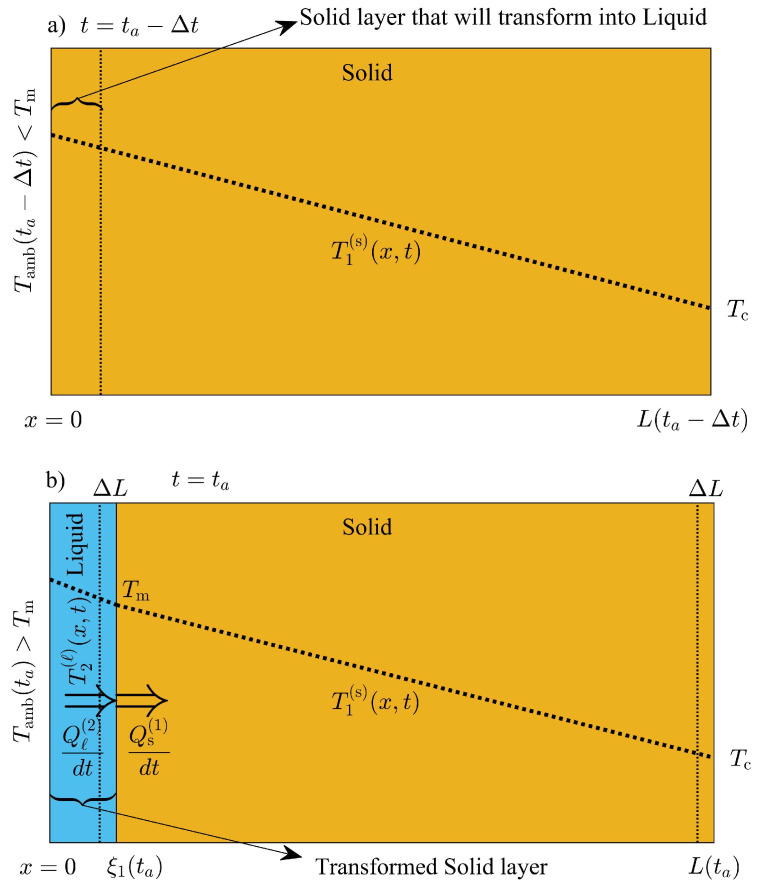
Liquid phase formation close to the exterior surface produced by the change of the ambient temperature from $T_{\mathrm{amb}}\left(t_{a}-\Delta \;t\right)<T_{m}$ to $T_{\mathrm{amb}}\left(t_{a}\right)>T_{m}$ during a small time interval Δt. (**a**) Thin solid layer that will melt close to the exterior surface. The layer is still in its solid form at $t=t_{a}-\Delta \;t$, when the temperature on the exterior surface is just below $T_{m}$. (**b**) Volume expansion $\Delta \;L=L\left(t_{a}\right)-L\left(t_{a}-\Delta \;t\right)$ after the formation of the liquid layer. The exterior surface is pushed rightwards, due to the expansion of the solid layer after the transition to its liquid state.

**Figure 5 molecules-27-02158-f005:**
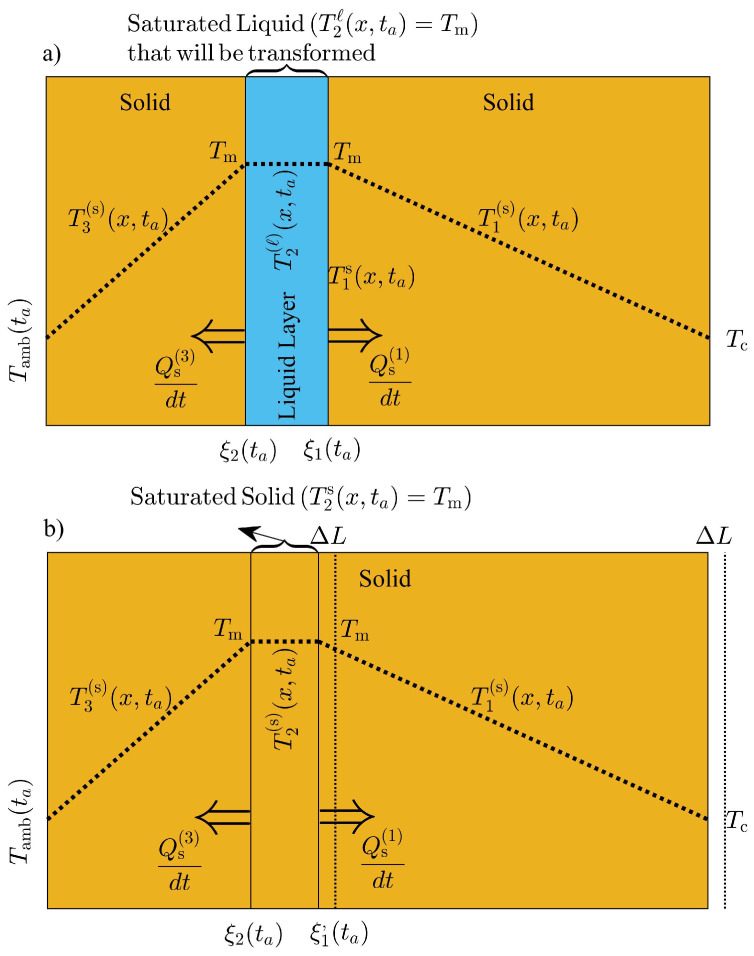
Solid phase formation during the collision of $\xi_{1}\left(t\right)$ and $\xi_{2}\left(t\right)$. (**a**) Saturated liquid layer before the phase transition at some time $t=t_{a}$. (**b**) PCM layer contraction $\Delta \;L=L\left(t_{a}\right)-L^{{}^{\prime }}\left(t_{a}\right)$ after the phase transition. The exterior surface is pulled to the left due to volume changes produced by the difference between solid and liquid densities.

**Figure 6 molecules-27-02158-f006:**
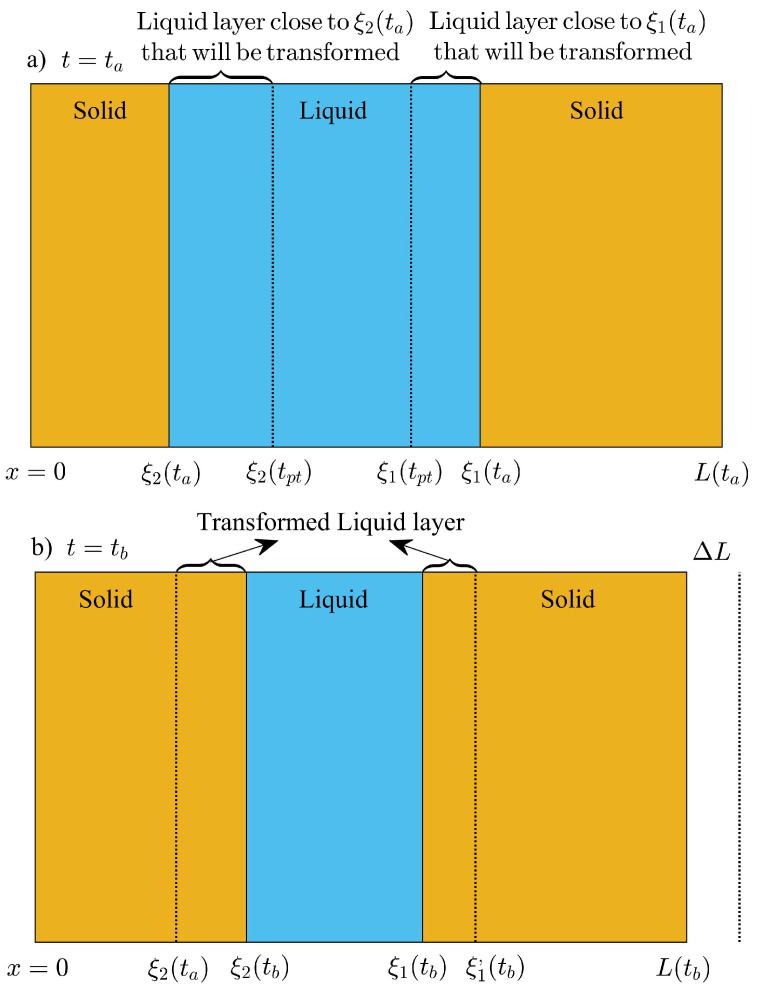
Schematic illustration of liquid layers $\xi_{2}\left(t_{pt}\right)-\xi_{2}\left(t_{a}\right)$ and $\xi_{1}\left(t_{a}\right)-\xi_{1}\left(t_{pt}\right)$, before and after the phase transition. (**a**) Saturated liquid layers before the transformation at some time $t=t_{a}$. The thickness of each liquid layer $\xi_{2}\left(t_{pt}\right)-\xi_{2}\left(t_{a}\right)$ and $\xi_{1}\left(t_{a}\right)-\xi_{1}\left(t_{pt}\right)$ is obtained by applying mass conservation. (**b**) Transformed liquid layers after the phase transition at some time $t=t_{b}$. The thickness of each solid layer $\xi_{2}\left(t_{b}\right)-\xi_{2}\left(t_{a}\right)$ and $\xi_{1}^{{}^{\prime }}\left(t_{b}\right)-\xi_{1}\left(t_{b}\right)$ is determined through mass conservation.

**Figure 7 molecules-27-02158-f007:**
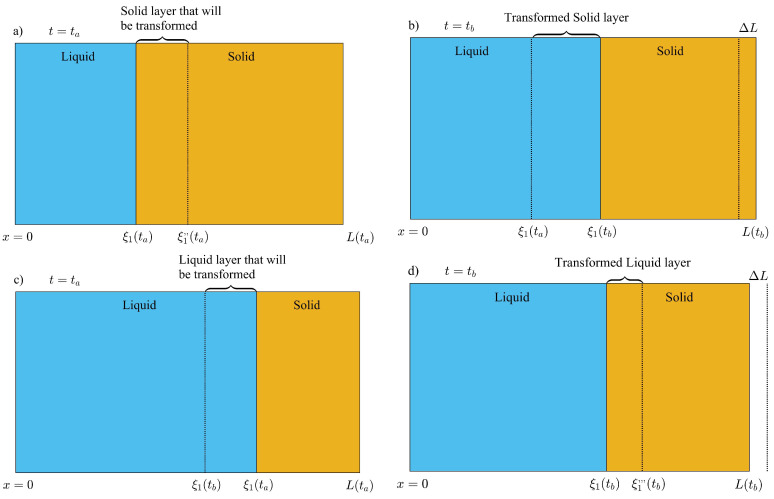
Volume changes during melting and solidification in a PCM layer with a two phase configuration (one-front). (**a**) Saturated solid layer in direct contact with $\xi_{1}\left(t_{a}\right)$ that will transform into its liquid phase. (**b**) Transformed solid layer during the melting process between $t_{a}$ and $t_{b}$. (**c**) Liquid layer at some time $t=t_{a}$ that will change to its solid phase. (**d**) Transformed liquid layer in contact with $\xi_{1}\left(t_{b}\right)$ after the phase transition.

**Figure 8 molecules-27-02158-f008:**
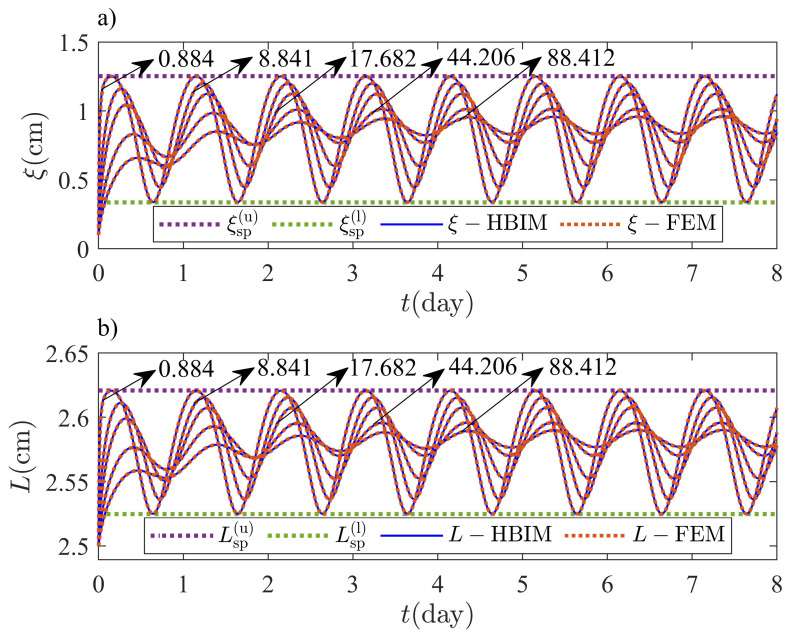
Transient and steady periodic behavior of $\xi_{1}\left(t\right)$ and L(t), and for several values of the Stefan number. Solid and dashed lines represent HBIM and FEM solutions, respectively for: (**a**) $\xi_{1}\left(t\right)$ and (**b**) layer thickness L(t). The amplitude of the oscillations in $\xi_{1}\left(t\right)$ and L(t) are tuned through the Stefan number by changing the value of $L_{f}$. Corresponding values of the Stefan number are indicated through black arrows. Lower and upper bounds in the steady periodic regime given by Equations (Equation 5)–(Equation 8) are shown through horizontal dotted lines.

**Figure 9 molecules-27-02158-f009:**
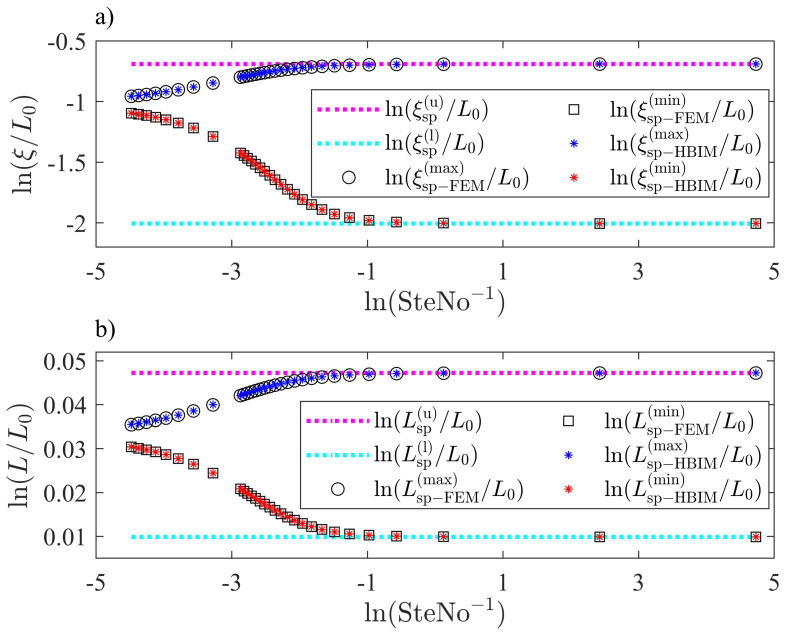
(**a**) HBIM and FEM steady periodic solutions as a function of the natural logarithm of the inverse Stefan number $\ln\left({\mathrm{SteNo}}^{-1}\right)$ for $\ln\left(\xi_{1}^{\left(\max\right)}\left(t\right)/L_{0}\right)$ and $\ln\left(\xi_{1}^{\left(\min\right)}\left(t\right)/L0\right)$, and (**b**) $\ln\left(L^{\left(\max\right)}\left(t\right)/L_{0}\right)$ and $\ln\left(L^{\left(\min\right)}\left(t\right)/L_{0}\right)$. Maxima and minima in $\xi_{1}\left(t\right)$ and L(t) were registered close to the steady periodic regime at t=9 days.

**Figure 10 molecules-27-02158-f010:**
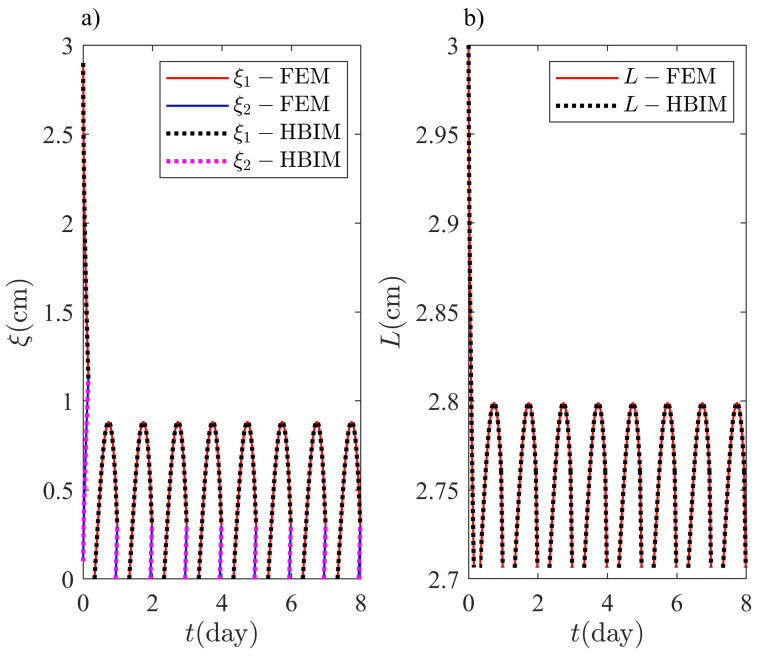
Time evolution of: (**a**) $\xi_{1}\left(t\right)$, $\xi_{2}\left(t\right)$, and (**b**) L(t), according to the HBIM and FEM solutions to Equations (Equation 4), (Equation 13), (Equation 16), and (Equation 18). Solid and dotted lines represent the FEM and HBIM solutions, respectively.

**Figure 11 molecules-27-02158-f011:**
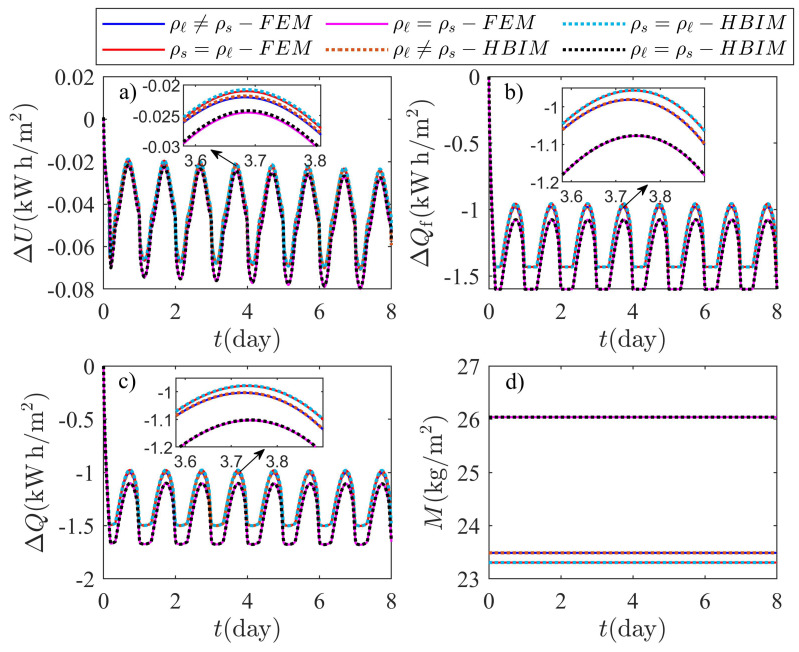
Time evolution of (**a**) internal energy change ΔU, (**b**) latent heat released (absorbed) $\Delta \;Q_{f}$, and (**c**) total energy changes ΔQ per unit area in $\mathrm{kWh}/m^{2}$ according to the model proposed in this work ($\rho_{s}\neq \rho_{\ell }$) and the constant volume method used by other authors ($\rho_{s}=\rho_{\ell }$ or $\rho_{\ell }=\rho_{s}$) [21,23,24,25]. (**d**) Time evolution of total mass is registered to verify that mass is not created or destroyed during the transient and steady periodic regimes. The effects of volume changes on the thermal energy released (absorbed) during solidification, melting, front formation and collision are discussed in Section 3.

**Figure 12 molecules-27-02158-f012:**
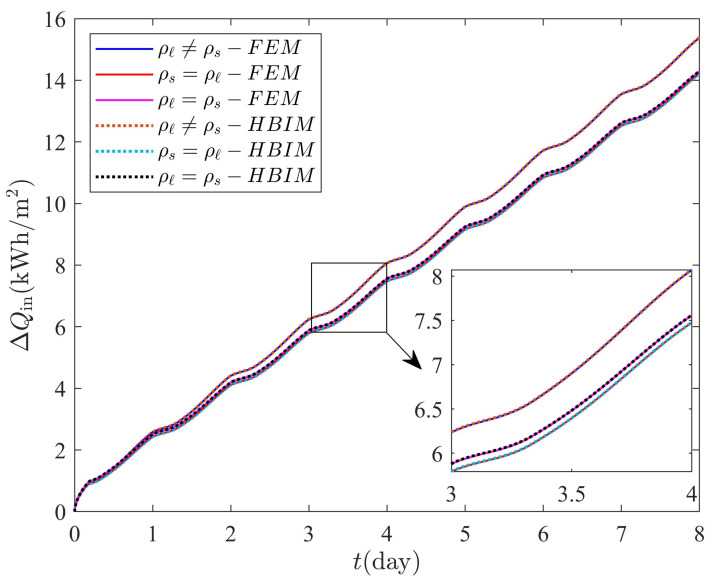
Energy transferred to the interior of the room according to the numerical and semi-analytical solutions used in this work. Constant volume methods of Refs. [21,23,24,25] are compared with the result obtained through the proposed model in this work. According to the initial state of the system and the result shown in this figure, the thermal performance of the PCM layer is enhanced by assuming equal densities.

**Table 1 molecules-27-02158-t001:** Maximum RPD for the sensible heat, latent heat and thermal energy released (absorbed) between the model discussed in this work and the constant volume methods used by other authors [21,23,24,25].

	${\mathrm{RPD}}_{\Delta \;U}^{\left(\max,1\right)}$	${\mathrm{RPD}}_{\Delta \;Q_{f}}^{\left(\max,1\right)}$	${\mathrm{RPD}}_{\Delta \;Q}^{\left(\max,1\right)}$	${\mathrm{RPD}}_{\Delta \;U}^{\left(\max,2\right)}$	${\mathrm{RPD}}_{\Delta \;Q_{f}}^{\left(\max,2\right)}$	${\mathrm{RPD}}_{\Delta \;Q}^{\left(\max,2\right)}$
FEM	3.73%	2.50%	2.50%	13.01%	11.72%	11.78%
HBIM	4.01%	2.51%	2.52%	12.84%	11.72%	11.77%

## Data Availability

The data presented in this study are available on request from the corresponding author.

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
