# Peer review of "Effects of Volume Changes on the Thermal Performance of PCM Layers Subjected to Oscillations of the Ambient Temperature: Transient and Steady Periodic Regimes"

_molecules, 2022, doi:10.3390/molecules27072158_

Round 1
Reviewer 1 Report
The authors present a thorough description of a model of PCM incorporating volume change. The technical treatment is complete, and there are only minor concerns with the model description and example data presentation. At this time I do have some issue with the narrative treatment around the model presentation, most specifically in how it builds upon previous work and conveys the rationale and impact for what is described. Details are provided below, and I hope that the suggestions can help improve the presentation of the hard work contained in the paper so that it's impact in the PCM modeling space can be fully realized.
- The introduction and literature review are comprehensive and address a wide range of the relevant research in the field. However, what is missing are statements liking the work that came before to the need to perform the work your paper describes. E.g., the 2nd paragraph in the introduction describing numerical simulation with volume changes states some details of previous work, but fails to state what is still missing from this area, what problems are still insufficiently solved, what is yet to have been done, and, most importantly, how it is that your work is attempting to improve upon that work or fill a still existing research gap. You end the introduction with an explanation of your overall modeling approach, but you missed explaining why that approach was taken, and how it relates to (or attempts to correct) that which came before. this needs to be corrected to place your work's impact in proper context. Why wasn't previous work sufficient and how will yours address that.
- I would argue that you also need to explain the need for this work - has previous research, experiment, or practice demonstrated a significant problem stemming from the lack of volume change in commonly used models? and how signifiacnt has that problem been shown to be? (i.e., help justify the importance of the problem to be solved)
- Specific to the modeling, you state clearly how you will be going about the model, (and then repeat some of this throughout the model description, somewhat redundantly). But you fail to state your reasoning for many of the modeling decisions, nor do you address assumptions made in your model selection and how severely those limit applicability of the model. (examples - density constant with temperature, only changing with phase, lack of supercooling and related thermal gradient to drive your phase front, stress free/unrestricted end expansion, etc. ). Why this approach? Why not another one? What decisions draw on prior research relevations? Please provide rationale to guide the reader.
- Once issue with the introduction/rationale is corrected, please ensure that the conclusions circle back to this same rationale, to explain how well the work did in filling that research gap, what remains to be done? Are any assumptions still problematic enough to warrant addressing in later work? (Such as how this one is addressing prior work "no volume change on phase change" assumptions?), and whether any additional insights were gained. E.g., the last sentence is the closest to fulfilling this goal, and is the most signifiant in an otherwise lackluster conclusion (everything else is just restating what you did, which isn't necessary)
- that last statement identifies an apparent deficiency in non-volume changing models. Was this unexpected? Was it seen previously in other models? Was this discrepancy seen between previous models and experiment to suggest that this was the cause, and that this papers result is 'real' as opposed to a modeling artifact? Does this have the potential to solve the problem of phase change modeling not being accurate? will that problem be worse for different materials with different properties or under different conditions? This is your opportunity to stress the importance of your work, and to justify the work yet to come. Use it!
- The results show excellent agreement between HBIM and the FEM implementation of the developed model, and they show discrepancy with models that don't account for volume change. That said, it is hard to verify how well this model actually tracks reality, without an attempt to emulate real world data that exhibited noticeable effects of volume change. If a proof set of data exists that could be briefly compared to your model and prior models, it would be a highly valuable addition. if not, it should be mentioned in the conclusions as an acknowledged limitation of the work and set as necessary future work before the model can be usable for practical system modeling/simulation/design.
minor points:
- p4 lines numbered 122-123: a gradient cannot be small relative to a tempearture. perhaps you meant to say the the end gradients are small relative to the gradient throughout the PCM sections?
- Note that the figures comparing this data to models from previous work must have more explicit legends stating which traces are from which studies, and giving reference numbers.
Reviewer 2 Report
The effect of volume changes were incorporated through an equation of motion for the thickness of the system in this work. The modified equation of motion for each interface was obtained by coupling mass conservation with a local energy-mass balance at each front. Hence a volumetric corrections to the sensible and latent heat released/ absorbed) were introduced. The article is well organized and written well with adequate scientific rigor. However the following points may be addressed before acceptance.
- A discussion on boundary conditions at which single, two or multiple fronts appear ( to support the illustrations provided in Figs. 1 to 6)
- A quantitative comparison in terms of percentage improvement in prediction accuracy of the proposed equation compared to HBIM and FEM without accounting volume changes may be discussed and summarized in the conclusions section.
